# LiveOIBench: Can Large Language Models Outperform Human Contestants in Informatics Olympiads?

## Abstract

Competitive programming problems increasingly serve as valuable benchmarks to evaluate the coding capabilities of large language models (LLMs) due to their complexity and ease of verification. Yet, current coding benchmarks face limitations such as lack of exceptionally challenging problems, insufficient test case coverage, reliance on online platform APIs that limit accessibility. To address these issues, we introduce LiveOIBench, a comprehensive benchmark featuring 403 expert-curated Olympiad-level competitive programming problems, each with an average of 60 expert-designed test cases. The problems are sourced directly from 72 official Informatics Olympiads in different regions conducted between 2023 and 2025. LiveOIBench distinguishes itself through four key features: (1) meticulously curated high-quality tasks with detailed subtask rubrics and extensive private test cases; (2) direct integration of elite contestant performance data to enable informative comparison against top-performing human; (3) planned continuous, contamination-free updates from newly released Olympiad problems; and (4) a self-contained evaluation system facilitating offline and easy-to-reproduce assessments. Benchmarking 34 popular general-purpose and reasoning LLMs, we find that GPT-5 achieves a notable 81st percentile, a strong result that nonetheless falls short of top human contestant performance, who usually place above 90th. In contrast, among open-weight reasoning models, GPT-OSS-120B achieves only a 60th percentile, underscoring significant capability disparities from frontier closed models. Detailed analyses indicate that robust reasoning models prioritize precise problem analysis over excessive exploration, suggesting future models should emphasize structured analysis and minimize unnecessary exploration. We have made the code of our benchmark accessible at `https://anonymous.4open.science/r/LiveOIBench-25F9/`.

## 1 Introduction

Coding has emerged as a critical domain for LLMs (Zhuo et al., 2024; Lai et al., 2022; Liu et al., 2024; Jimenez et al., 2024; Chan et al., 2024), with coding benchmarks serving as essential tools to evaluate LLMs' algorithmic reasoning capabilities as these models continue advancing through inference-time scaling techniques (Li et al., 2022a; Kojima et al., 2023; DeepSeek-AI et al., 2025; OpenAI et al., 2024; Li et al., 2025b). However, rapid improvements in model capabilities have led to saturation of traditional coding benchmarks such as HumanEval (Chen et al., 2021a) and MBPP (Austin et al., 2021), prompting the adoption of competitive coding benchmarks (Li et al., 2022a; Hendrycks et al., 2021b; Li et al., 2023; Shi et al., 2024) such as LiveCodeBench (Jain et al., 2024) and CodeELO (Quan et al., 2025), which leverage problems from platforms like Codeforce for their complexity and ease of verification. Despite their strengths, these benchmarks have notable weaknesses: (1) overestimation of LLMs' performance due to high false-positive rates using incomplete test suites (Li et al., 2022a; Liu et al., 2023; Jain et al., 2024), (2) insufficient difficulty granularity and lacking exceptionally challenging questions (Jain et al., 2024; Quan et al., 2025), (3) usage of external APIs for evaluation, restricting reproducibility and accessibility (Jain et al., 2024; Quan et al., 2025; Zheng et al., 2025; Li et al., 2025c), (4) reliance on coarse pass rates as the sole evaluation metric, which misses the opportunity to gain insights on nuanced model capabilities (Jain et al., 2024; Li et al.,

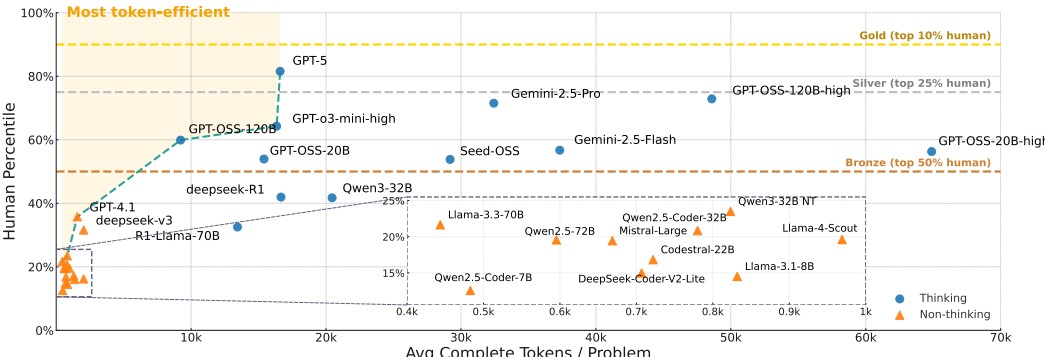

Figure 1: **LiveOIBench.** Average human percentile across all contests versus average completion tokens per problem. The dashed boxes highlight the lower performance range of non-thinking LLMs. OpenAI models lie on the token-efficiency frontier, achieving higher human percentile with fewer tokens. Despite improvements, all evaluated models remain below the Gold medal threshold (top 10% human performance), indicating substantial room for progress.

2022a; Wang et al., 2025a; Shi et al., 2024), and (5) infrequent or costly updates due to the extensive human annotations and computational resources required (Wang et al., 2025a; Zhu et al., 2025).

To address these gaps, we introduce LiveOIBench, the first comprehensive Informatics Olympiads coding benchmark constructed directly from official contest sources, featuring expert-designed private tests and official contestant rankings, which will be made publicly available to support reproducible evaluation along with fine-grained scoring rubrics. Compared to previous benchmarks (Jain et al., 2024; Shi et al., 2024; Hendrycks et al., 2021b; Li et al., 2022a; Quan et al., 2025) and concurrent work (Li et al., 2025c; Zheng et al., 2025; Zhu et al., 2025; Wang et al., 2025a) in Table 1, LiveOIBench features the following key advancements:

1. **Expert-curated Tasks with Fine-grained Subtask Rubrics.** We curate problems, test cases, and scoring rubrics directly from the official websites of 14 Informatics Olympiads. This comprehensive test suite eliminates high false-positive rates common in previous benchmarks (Li et al., 2022a; Liu et al., 2023; Jain et al., 2024). Additionally, each task includes subtasks with scoring rubrics, enabling nuanced insights into model capabilities.

2. **Direct Human Contestant Comparisons.** Official results from top human competitors are collected, allowing direct and informative benchmarking against human-level performance.

3. **Continuous, Contamination-free Updates.** Updates with newly released Olympiad tasks maintain benchmark freshness and minimize data contamination risks, supporting continuous monitoring of LLM coding capabilities on challenging programming problems.

4. **Integrated Offline Evaluation System.** We develop a self-contained evaluation judge, enabling fully offline and reproducible model evaluation without relying on external APIs or online platforms, significantly enhancing accessibility and reproducibility.

In total, LiveOIBench comprises 403 rigorously curated problems sourced from 72 contests across 14 Informatics Olympiads, each accompanied by an average of 60 expert-written test cases. Using LiveOIBench, we evaluate 34 leading models, revealing that proprietary models maintain a substantial performance advantage. In particular, GPT-5 (OpenAI, 2025b) achieves an average human percentile of 82, while also exhibiting remarkable token efficiency by reaching this performance with fewer than 20K reasoning tokens, positioning it on the efficiency frontier (Figure 1). Among open-weight alternatives, Seed-OSS (ByteDance Seed Team, 2025) achieves the 54th percentile and Qwen-3-32B (Yang et al., 2025b) reaches the 42nd percentile, both demonstrating significant performance gains from additional reasoning tokens. Additionally, GPT-OSS-120B (OpenAI et al., 2025) attains the 60th percentile, effectively narrowing the performance gap with GPT-5 and highlighting significant progress in open-weight model capabilities. Moreover, examining performance across different algorithms reveals current models' weaknesses in algorithms like dynamic programming, which demand creative observation and hierarchical reasoning. Additionally, detailed reasoning trace analyses reveal that high-performing models strategically allocate more tokens to focused analysis

| Dataset | Difficulty | Updates | Expert Test Cases | Offline Eval | Subtasks | Human Percentile |
|---|---|---|---|---|---|---|
| HumanEval | ★ | ✗ | ✗ | ✓ | ✗ | ✗ |
| APPS | ★★ | ✗ | ✗ | ✓ | ✗ | ✗ |
| CodeContests | ★★★ | ✗ | ✗ | ✓ | ✗ | ✗ |
| TACO | ★★ | ✗ | ✗ | ✓ | ✗ | ✗ |
| LiveCodeBench | ★★ | ✓ | ✗ | ✓ | ✗ | ✗ |
| USACO | ★★★ | ✓ | ✓ | ✓ | ✗ | ✗ |
| CODEELO | ★★★ | ✓ | ✓(hidden) | ✗ | ✗ | ✓ |
| OI-Bench | ★★★★ | ✗ | ✓(unofficial) | ✗ | ✗ | ✓ |
| LiveCodeBench-Pro | ★★★★ | ✓ | ✓(hidden) | ✗ | ✗ | ✓ |
| HLCE | ★★★★ | ✗ | ✓(hidden) | ✗ | ✗ | ✓ |
| AetherCode | ★★★★ | ✓ | ✓ (unofficial) | ✗ | ✗ | ✗ |
| LiveOIBench (Ours) | ★★★★ | ✓ | ✓✓ | ✓ | ✓ | ✓ |

Table 1: Comparison with existing coding datasets. LiveOIBench consists of continuously updated competitive coding problems from recent Informatics Olympiads, spanning various difficulty levels. Unlike previous benchmarks that generated test cases using predefined rules or LLMs, LiveOIBench features expert-curated *private* test cases sourced *directly* from official competition organizers. It also provides an accessible *offline* evaluation platform, detailed subtask rubrics for *fine-grained* assessment, and official human contestant rankings for precise *human-model* comparisons.

rather than excessive exploration, underscoring that carefully managed reasoning behaviors are crucial for robust performance on challenging tasks.

In summary, we make the following key contributions:

- **(Data)** Curate and release a comprehensive, high-quality competitive coding benchmark with expert-crafted problems, hidden test suites, and integrated human contestant results.
- **(Evaluation)** Provide a robust local evaluation framework with private test cases and detailed subtask scoring rubrics, enabling accessible, fine-grained human-model comparisons.
- **(Benchmarking Results)** Conduct extensive benchmarking and detailed performance analysis of 32 leading open-source and proprietary models.
- **(Analyses)** Perform extensive analyses such as evaluating model performance across diverse algorithms, detailed reasoning trace analyses, examination of solution submission outcomes, and assessments of model performance under inference-time scaling.

## 2 RELATED WORK

The early code generation benchmarks HumanEval (Chen et al., 2021a) and MBPP (Austin et al., 2021) mainly focus on the basic Python programs, which, for a long time, have been the standard ways to evaluate the code generation capability of LLMs. However, as the capability of LLMs evolves, simple benchmarks like HumanEval can no longer satisfy the benchmarking needs. Researchers have started developing more realistic and challenging benchmarks (Zhuo et al., 2024; Lai et al., 2022; Liu et al., 2024; Jimenez et al., 2024; Chan et al., 2024; Yin et al., 2023). Specifically, DS1000 (Lai et al., 2022) and ARCADE (Yin et al., 2023) consist of data science problems in Python. BigCodeBench (Zhuo et al., 2024) collects code generation tasks from Stack Overflow, which involves more complex instructions and diverse function calls. The SWE-Bench (Jimenez et al., 2024) takes one step further and tests models' ability to solve real-world Github issues. This line of work emphasizes evaluating LLMs' ability to effectively implement, debug, and reason through complex real-world coding tasks.

In addition to real-world application benchmarks, there is another line of work: **competitive programming benchmarks** (Li et al., 2022a; Hendrycks et al., 2021b; Jain et al., 2024; Quan et al., 2025), which test the reasoning ability of models to solve challenging coding tasks within the specified time and memory constraints. All the previous competitive programming benchmarks collect problems from online coding platforms like Codeforces, which do not release private test cases. The lack of sufficient private test cases may cause many false-positive solutions (Li et al., 2022a; Liu et al., 2023). Li et al. (2022a) augment test cases by mutating existing test inputs. Liu et al. (2023) leverages both LLM-based and mutation-based strategies to augment test cases with predefined rules. Even with over 200 additional tests per problem, Li et al. (2022a) shows there still exists nearly 50% false-positive rates. Other work (Quan et al., 2025; Zheng et al., 2025; Li et al., 2025c) tries to solve this problem by creating a platform to submit LLM-generated solutions directly

to the Codeforces platform. Although this approach ensures that solutions are tested on the whole test set, its dependency on the online platform limits its accessibility to the research community, as large-scale evaluations involving thousands of submissions can overload platform servers.

To solve the above problem, we collect problems from the official websites of many informatics Olympiads around the world. Most informatics Olympiads release their complete test set, which is curated carefully by the organizing committees. We are one of the first works to leverage problems from different informatics Olympiads and evaluate the models' performance against human contestants. Prior research by Shi et al. (2024) exclusively used USACO problems with pass rate as the sole evaluation metric. Concurrent benchmarks, such as LiveCodeBench Pro (Zheng et al., 2025), HLCE (Li et al., 2025c), OI-Bench (Zhu et al., 2025), and AetherCode (Wang et al., 2025a), also incorporate competitive programming tasks from sources like ICPC and IOI. However, LiveCodeBench Pro and HLCE primarily evaluate using Codeforces, limiting their accessibility. OI-Bench relies mostly on private, non-English school contests without continuous updates, while AetherCode uses LLM-generated tests and extensive human annotation with pass rate evaluation only. In contrast, our benchmark provides comprehensive coverage across diverse Olympiads, allows easy updates by directly collecting official test cases, and employs detailed evaluation metrics including subtask rubrics and human percentile comparisons.

## 3 LIVEOIBENCH CONSTRUCTION

To construct LiveOIBench, we follow a clearly defined, step-by-step process combining automated data collection methods with manual verification to ensure dataset quality.

**Competition Selection and Task Collection:** We first curate a comprehensive list of globally recognized international Informatics Olympiads and selectively incorporate national contests from top-performing IOI countries where English task statements are available (See Table A5). For each selected contest, we develop a custom crawler that systematically extracts English task statements (See Appedix B.5) directly from official competition websites, capturing details such as time and memory constraints, subtask specifications, test cases, official solutions, and contestant rankings. When official sites lack complete or up-to-date information, we supplement the data by retrieving missing details from established online platforms such as CSES[1] and LibreOJ[2]. To mitigate potential contamination from pre-training datasets, we strictly limit our dataset to contests held after 2022. Additionally, we provide full descriptions of each competition along with official websites in Appendix B.6, ensuring selected contests have extensive historical data, consistent participant numbers, and regularly hosted events. Our benchmark will be continuously updated with new contests and maintain an active leaderboard using a website similar to Figure A1.

**Markdown Conversion and Quality Assurance:** Given that many contests provide task statements exclusively as PDF documents, we employ Marker[3] to automatically convert these PDFs into markdown format. We further utilize Gemini-2.0-Flash to automatically verify and correct these markdown texts. To ensure conversion accuracy, we manually inspect a sample of 40 tasks before batch processing. Additionally, we verify our evaluation judge and crawled test cases by executing the official solutions from contest organizers, using these solutions as the ground truth to confirm test-case correctness and the robustness of our evaluation judge.

**Metadata Enrichment:** We enhance the dataset with supplementary metadata, including difficulty and algorithm tags, crawled from solved.ac[4] and Luogu[5]. Tasks and metadata are matched using competition dates, task titles, and problem identifiers. More details can be found in Appendix B.3.

**Contestant Matching and Codeforces Ratings:** Beyond raw human contestant results, contestants are automatically linked to their respective Codeforces profiles based on their names, user IDs, and countries, while contestants whose profiles cannot be confidently matched are skipped. Verified profiles are then queried via the Codeforces API to retrieve user ratings from 2022 to 2025. More details can found in Appendix B.4 and Table A6.

---

[1] https://cses.fi
[2] https://loj.ac
[3] https://github.com/datalab-to/marker
[4] https://solved.ac
[5] https://www.luogu.com.cn

| Model | Gold (%) | Medals (%) | Relative Score(%) | Human Percentile (%) | Pass Rate (%) | ELO |
|---|---|---|---|---|---|---|
| **Proprietary LLMs** | | | | | | |
| GPT-5 | **50.00** | **88.89** | **67.21** | **81.76** | **63.03** | **2414** |
| Grok-4-Fast-Reasoning | 45.83 | 83.33 | 56.99 | 74.23 | 50.95 | 2221 |
| Gemini-2.5-Pro | 31.94 | 77.78 | 51.33 | 71.80 | 44.46 | 2192 |
| GPT-O3-Mini-High | 26.39 | 72.22 | 47.69 | 64.28 | 44.19 | 2088 |
| Gemini-2.5-Flash | 15.28 | 62.5 | 41.29 | 56.81 | 36.06 | 1945 |
| Claude-Sonnet-4.5 | 11.11 | 66.68 | 38.30 | 53.08 | 27.05 | 1848 |
| GPT-4.1 | 4.17 | 40.28 | 24.78 | 35.99 | 18.32 | 1482 |
| **Open-weight Thinking LLMs** | | | | | | |
| GPT-OSS-120B-High | **50.00** | 87.50 | 62.78 | 72.88 | 60.14 | 2205 |
| GPT-OSS-120B | 29.17 | 73.61 | 49.23 | 59.90 | 47.78 | 2032 |
| GPT-OSS-20B | 19.44 | 68.06 | 42.36 | 53.94 | 42.80 | 1901 |
| Seed-OSS | 15.28 | 68.06 | 42.58 | 53.81 | 40.09 | 1873 |
| Qwen3-32B | 9.72 | 54.17 | 32.86 | 42.00 | 27.70 | 1665 |
| DeepSeek-R1 | 6.94 | 52.78 | 33.43 | 42.29 | 28.87 | 1617 |
| Qwen3-14B | 5.56 | 45.83 | 27.24 | 34.59 | 22.73 | 1402 |
| DeepSeek-R1-Distill-Llama-70B | 1.39 | 33.33 | 20.50 | 32.30 | 16.88 | 1284 |
| **Open-weight Non-Thinking LLMs** | | | | | | |
| DeepSeek-V3 | 4.17 | 34.72 | 21.70 | 31.76 | 17.10 | 1283 |
| Qwen3-32B-Non-Thinking | 1.39 | 16.67 | 12.92 | 24.64 | 8.78 | 1040 |

Table 2: Main results of best-performing models in each category evaluated on all 72 contests. Full results are presented in Table A9. **Gold** and **Medals**: % of contests in which a model achieved a gold medal or any medal, respectively. **Relative Score**: % of total contest points obtained by the model. **Human Percentile**: % of human contestants that a model surpasses. **Pass Rate**: % of tasks where a model successfully passes all test cases. **ELO**: the Codeforces ELO rating earned by a model based on performance relative to human contestants. All metrics are higher the better. Notably, the highest-performing GPT-5 achieves an impressive 81.76 percentile but still falls short of top human contestants, successfully solving only 63% of tasks in the benchmark.

Ultimately, LiveOIBench comprises 403 **rigorously curated problems** from 72 **competitions** across 14 **Informatics Olympiads**, conducted between 2023 and 2025. The benchmark statistics are detailed in Table A4, with a detailed description of our dataset construction methodology provided in Appendix B and competition information in Appendix B.6.

There are four characteristics that make our dataset challenging and unique compared to the existing coding datasets:

- **Challenging Problems with Subtasks.** Expert-curated problems contain subtasks with distinct constraints, enabling precise evaluation through partial scoring.

- **Expert-Designed Private Tests.** Includes expert-designed private tests rather than test cases generated by predefined rules or LLMs, ensuring evaluation free of false positives.

- **Direct Human Comparisons.** Benchmarks LLM performance against human contestants using percentile ranks, medals, and Codeforces ELO ratings.

- **Live Updates.** Continuously updated with recent contests to minimize data contamination. All 14 competitions described in Appendix B.6 in our benchmark will be updated.

## 4 BENCHMARKING RESULTS

We evaluate a comprehensive set of 34 LLMs. These models are categorized into three groups based on their accessibility and "thinking" capabilities: proprietary LLMs, open-weight thinking LLMs, and open-weight non-thinking LLMs. More details about models can be found Appendix C. During inference, we sample 8 candidate solutions per model in C++ and pick the solution with the highest score (Jain et al., 2024; Quan et al., 2025). We adopt following evaluation metrics: pass rate (Kulal et al., 2019; Chen et al., 2021b), relative score, human percentile, Olympics medal system, and Codeforces ELO (Quan et al., 2025; Zheng et al., 2025). With subtask rubrics and human contestants results, we can calculate each model's total points in a contest, allowing precise comparisons to human contestants via percentile rankings and medal awards. The description of each metric can be found in Table 2 or Appendix D. In Table 2, we present benchmarking results for selected models. Full results for all evaluated models are included in Table A9.

**Proprietary LLMs remain dominant, yet open-weight models are narrowing the performance gap.** Our findings indicate that proprietary LLMs continue to lead in competitive coding benchmarks. Specifically, GPT-5 achieves impressive results, securing gold medals in $50\%$ of contests, winning medals of any type in $88.89\%$ of contests, and outperforming an average of $81.76\%$ of human contestants. Among open-source models tested, GPT-OSS-120B emerges as the strongest competitor. Under standard reasoning effort, GPT-OSS-120B achieves gold medals in $29.17\%$ of contests and performs near the 60th percentile—approximately 21.86 percentile points below GPT-5. Notably, with high reasoning effort, GPT-OSS-120B surpasses Gemini-2.5-Pro and trails GPT-5 by merely 9 percentile points. Seed-OSS, the second-best open-source model, attains the 54th percentile, narrowly trailing Gemini-2.5-Flash by only 3 percentile points. However, other models exhibit substantial performance gaps, with Qwen3-32B and Deepseek-R1 obtaining gold medals in only $10\%$ and $7\%$ of contests, respectively, and performing at roughly the 42nd percentile.

**Even the leading GPT-5 model falls short of top-tier human contestants.** Achieving a gold medal in every contest requires consistently surpassing the 90th percentile. Although GPT-5 demonstrates remarkable capabilities with a near 82nd percentile and a rating of 2414, its performance still lags behind elite human competitors. This highlights an ongoing challenge for LLMs in surpassing human expertise in competitive coding.

**Thinking models perform significantly better than non-thinking models.** Models lacking extended thinking capabilities perform notably worse in our benchmark. GPT-4.1, the highest-performing non-thinking model evaluated, achieves results comparable only to Qwen3-14B. Apart from Deepseek-V3, all other non-thinking models fail to exceed a $10\%$ pass rate, underscoring the critical importance of extended thinking in addressing complex competitive coding tasks. Extending this analysis, we investigate inference-time scaling techniques and find that both parallel (Chen et al., 2021b; Jain et al., 2024) and sequential (DeepSeek-AI et al., 2025; Snell et al., 2024; Li et al., 2025a) scaling methods significantly enhance coding capabilities. In Figure 2, parallel scaling identifies maximum coding capacity but shows diminish-

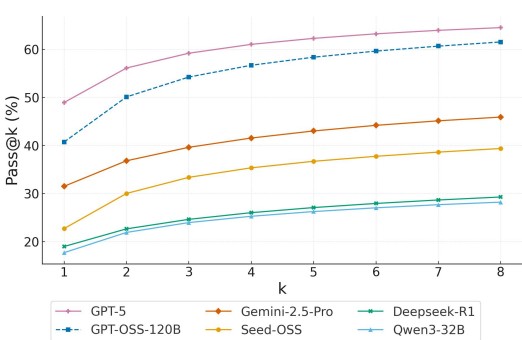

Figure 2: **Parallel Scaling** displays the Pass@k performance, illustrating how the success rate improves as more solutions (k) are sampled per problem. GPT-5 shows the highest sample efficiency and overall performance ceiling.

ing returns beyond a few attempts, while sequential scaling, by increasing the reasoning budget, allows smaller models to approach larger-model performance in Figure A4, reinforcing our earlier observation on the importance of extended thinking capabilities. For detailed analyses, see Appendix F.2.

**Comprehensive evaluation metrics provide deeper insights into model capabilities.** Relying solely on pass rate can obscure key aspects of model performance. For example, GPT-OSS-120B achieves a higher pass rate ($47.78\%$) compared to Gemini-2.5-Pro ($44.46\%$); however, Gemini-2.5-Pro consistently surpasses GPT-OSS-120B in both human percentile ranking and ELO rating, indicating stronger overall competitive coding proficiency. We recommend that practitioners and researchers adopt a multifaceted evaluation approach: use Gold and Medals to gauge contest-level success, Human Percentile to contextualize model performance relative to humans, ELO to assess coding skill within the broader competitive coding community, and Pass Rate to evaluate core problem-solving capability. Utilizing these metrics collectively ensures a balanced and comprehensive understanding of model strengths and limitations.

**Later subtasks are more challenging.** We investigate how model performance is affected by the sequential position of subtasks within problems. Specifically, we segment all subtasks into five equal bins based on their relative positions and observe a consistent decline in model performance for subtasks appearing later in the sequence, as illustrated in Figure A2. This result is intuitive, as earlier subtasks typically impose stronger constraints on input variables, making them easier and prerequisites for subsequent subtasks. In contrast, later subtasks usually lack explicit constraints, requiring more generalized and optimized solutions.

| Model | IM | MA | AH | PS | SO | GR | GTR | BS | NT | GT | DS | CB | DP | TR | ST |
|---|---|---|---|---|---|---|---|---|---|---|---|---|---|---|---|
| 🔒 Proprietary LLMs | | | | | | | | | | | | | | | |
| GPT-5 | 71.79 | 71.43 | 43.48 | 73.33 | 75.56 | 60.00 | 71.43 | 54.84 | 64.71 | 66.67 | 66.27 | 64.71 | 46.88 | 37.50 | 56.41 |
| GEMINI-2.5-PRO | 66.67 | 71.43 | 30.43 | 53.33 | 57.78 | 37.14 | 42.86 | 38.71 | 35.29 | 44.44 | 38.55 | 58.82 | 23.44 | 20.83 | 30.77 |
| GPT-O3-MINI-HIGH | 64.10 | 71.43 | 34.78 | 46.67 | 60.00 | 37.14 | 46.43 | 41.94 | 41.18 | 38.89 | 38.55 | 47.06 | 34.38 | 20.83 | 28.21 |
| GEMINI-2.5-FLASH | 64.10 | 71.43 | 30.43 | 46.67 | 48.89 | 28.57 | 25.00 | 32.26 | 29.41 | 29.63 | 30.12 | 47.06 | 20.31 | 12.50 | 15.38 |
| GPT-4.1 | 53.85 | 50.00 | 26.09 | 40.00 | 13.33 | 14.29 | 7.14 | 12.90 | 17.65 | 12.96 | 12.05 | 29.41 | 6.25 | 4.17 | 5.13 |
| Open-source Thinking LLMs | | | | | | | | | | | | | | | |
| GPT-OSS-120B | 64.10 | 64.29 | 34.78 | 53.33 | 60.00 | 40.00 | 53.57 | 38.71 | 41.18 | 44.44 | 44.58 | 58.82 | 35.94 | 25.00 | 35.90 |
| GPT-OSS-20B | 63.16 | 71.43 | 40.91 | 57.14 | 51.11 | 36.36 | 35.71 | 36.67 | 47.06 | 30.19 | 36.59 | 66.67 | 29.69 | 22.73 | 26.32 |
| SEED-OSS | 61.54 | 64.29 | 36.36 | 53.33 | 48.89 | 31.43 | 32.14 | 38.71 | 35.29 | 27.78 | 34.94 | 52.94 | 26.56 | 12.50 | 28.21 |
| QWEN3-32B | 58.97 | 61.54 | 30.43 | 35.71 | 28.89 | 21.88 | 21.43 | 16.67 | 29.41 | 22.64 | 22.22 | 29.41 | 14.29 | 4.35 | 8.11 |
| DEEPSEEK-R1 | 61.54 | 64.29 | 30.43 | 33.33 | 28.89 | 17.14 | 17.86 | 22.58 | 29.41 | 22.22 | 20.48 | 29.41 | 15.62 | 4.17 | 7.69 |
| DEEPSEEK-R1-DISTILL-LLAMA-70B | 41.03 | 50.00 | 17.39 | 20.00 | 20.00 | 17.14 | 10.71 | 16.13 | 17.65 | 14.81 | 13.25 | 11.76 | 9.38 | 4.17 | 5.13 |
| Open-weight Non-Thinking LLMs | | | | | | | | | | | | | | | |
| DEEPSEEK-V3 | 51.28 | 46.15 | 21.74 | 28.57 | 20.00 | 12.50 | 14.29 | 13.33 | 17.65 | 15.09 | 14.81 | 11.76 | 7.94 | 8.70 | 8.11 |
| QWEN3-32B-NON-THINKING | 25.64 | 42.86 | 13.04 | 0.00 | 6.67 | 5.71 | 3.57 | 9.68 | 11.76 | 7.41 | 2.41 | 11.76 | 4.69 | 0.00 | 2.56 |

Table 3: Pass@8 of top-15 algorithm tags for selected models. Full results can be found in Table A10. Abbreviations: IM (implementation), MA (mathematics), AH (ad-hoc), PS (prefix sum), SO (sorting), GR (greedy), GTR (graph traversal), BS (binary search), NT (number theory), GT (graph theory), DS (data structures), CB (combinatorics), DP (dynamic programming), TR (tree), ST (segment tree). Darker color indicates the model performs better on this particular tag compared to other tags. Models generally perform better on algorithm tags that involve straightforward application of standard formulas or well-known patterns.

## 5 IN-DEPTH ANALYSES OF MODEL BEHAVIOR AND ERROR PATTERNS

We first analyze algorithmic complexity to identify models' strengths and weaknesses, then explore their strategic reasoning behaviors, and finally investigate specific error patterns to pinpoint areas for model improvement.

### 5.1 ALGORITHMIC COMPLEXITY DETERMINES MODEL PERFORMANCE PATTERNS

**Models are generally proficient at algorithm tags that require basic mathematical procedures and minimal compositional reasoning**. As shown in Table 3, all evaluated models consistently achieve higher pass rates on tasks categorized under implementation, mathematics, prefix sum, sorting, and graph traversal—GPT-5 notably attains over 70% accuracy on most of these tags. Such tasks primarily depend on recognizing familiar solution templates or leveraging procedural knowledge obtained from training. Performance noticeably declines for algorithms demanding deeper analytical reasoning or succinct proofs, such as greedy methods and graph theory, where even top proprietary models like GPT-5 drop to around 60%. The greatest difficulties arise in tasks that require on-the-spot creative observations, intricate state designs, or hierarchical invariants—particularly evident in dynamic programming (DP), segment trees (ST), and tree (TR) problems, where GPT-5's pass rate sharply decreases to approximately 47%, 56%, and 38%, respectively. *To address these weaknesses*, future work could explore curriculum-driven fine-tuning (Huang et al., 2025) using carefully designed synthetic datasets of complex graph, tree, and DP problems, encouraging models to internalize the recurrence relations, hierarchical invariants, and compositional reasoning patterns crucial to solving these more challenging algorithmic tasks.

### 5.2 REASONING TRACE ANALYSES: STRONGER MODELS ALLOCATE REASONING TOKENS MORE STRATEGICALLY

To better understand how thinking models solve challenging competitive coding problems, we conduct a detailed analysis on models' reasoning traces. Inspiring by prior work (Gandhi et al., 2025; Ahmad et al., 2025) on reasoning behavior analysis, we categorize models' reasoning traces into eight behaviors and classify them into five groups as shown in Figure 3. Each trace is segmented into shorter chunks and annotated using GPT-OSS-120B. More details on the annotation prompt and implementation can be found in Appendix F.8.

**GPT-OSS-120B increases exploration and analysis with problem difficulty, yet maintains stable exploration levels across reasoning budgets.** In Figure 3a, on more challenging problems, GPT-OSS-120B-High devotes significantly more effort to exploration—searching for viable solution paths—and deeper problem analysis, simultaneously reducing the tokens spent on initial planning

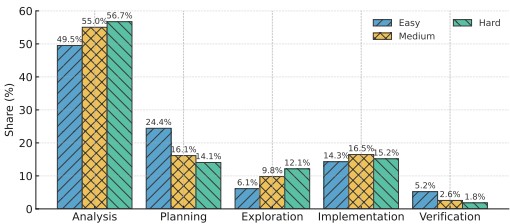
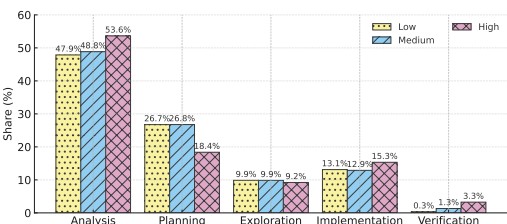

(a) GPT-OSS-120B-high across Easy/Medium/Hard. As problem difficulty increases, models prioritize exploration and analysis over planning and verification.

(b) GPT-OSS-120B across reasoning efforts. Higher reasoning budgets lead to deeper analysis, implementation, and verification without increased exploration.

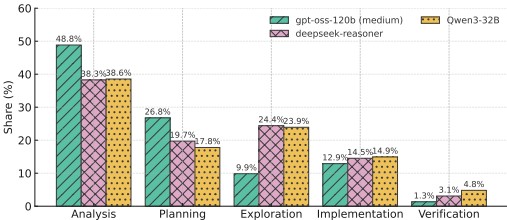
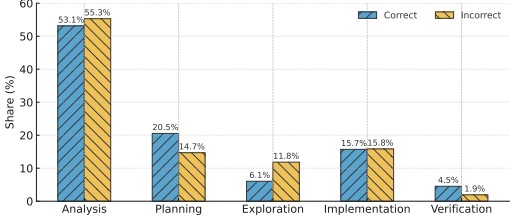

(c) Model comparison. Stronger reasoning models reduce unnecessary exploration, dedicating more resources to planning, structured analysis, and solution development.

(d) GPT-OSS-120B-high across correct/incorrect. Correct solutions depend heavily on initial structured planning and verification, reducing the need for exploration and continuous re-analysis.

Figure 3: **Reasoning Trace Analyses.** We categorized eight reasoning behaviors and divide them into five groups: **Analysis** (Algorithm/Proof analysis and Complexity Analysis), **Planning** (Problem Restatement and Subgoal Setting), **Exploration** (Backtracking and Dead-end recognition), **Implementation** (Pseudo implementation), **Verification** (Test Case Verification).

and verification. This indicates that initial problem structuring behaviors are typically conducted early and not revisited extensively once a potential solution path is identified. Notably, even when provided with increased reasoning budgets (from low to high reasoning effort), as shown in Figure 3b, GPT-OSS-120B strategically allocates extra tokens toward analysis, implementation, and verification, rather than further exploration. By maintaining stable exploration levels despite increased reasoning resources, the model mitigates excessive pivoting, a critical behavior that could otherwise lead to inefficient or incomplete reasoning traces, or "underthink" (Shojaee et al., 2025).

**Stronger reasoning models exhibit reduced exploration, allocating more resources toward solution development and analysis..** After problem difficulty and reasoning efforts, we further see, in Figure 3c and Figure A13, more capable models dedicate more reasoning tokens to problem understanding, structured planning, and detailed algorithmic analysis. Consequently, they spend less time pivoting to alternative paths, generating pseudo-implementations, or performing test-case verification. It highlights the future direction of effectively allocating models' problem analysis and exploration to avoid excessive pivoting and prevent "underthinking".

**Initial planning behaviors and subsequent verification steps play crucial roles in models producing correct solutions.** Building upon this observation, we also investigate which reasoning behaviors distinguish correct from incorrect solutions. As illustrated in Figure 3d, correct solutions exhibit increased planning behaviors, potentially explaining why exploration behaviors diminish—well-structured planning facilitates clearly defined solution paths, reducing the need for exploratory detours. Additionally, correct solutions engage in verification behaviors more frequently, ensuring adequate solution checks. This increased verification slightly reduces the need for extensive analysis, as models rely less on continuous reevaluation once confident in their solution correctness. Notably, correct solutions include more verification because targeted end-checks consolidate successful trajectories; however, stronger models rely less on explicit verification overall due to robust upfront analysis and planning, which internalize many checks and reduce the need for post-hoc verification.

## 5.3 ERROR PATTERNS IN MODEL-GENERATED CODE SUBMISSIONS

**Stronger reasoning capabilities in models correlate with reduced failure rates, yet runtime errors remain a notable challenge.** In Figure 4, we analyze the submission status distribu-

tion across six selected models. As models exhibit stronger reasoning capabilities, their solutions show substantial reductions in failure types of time limit, memory limit, and compilation errors. However, runtime errors, although somewhat reduced, do not experience as pronounced a decline, highlighting persistent challenges in edge-case handling and execution robustness.

We hypothesize that one possible reason top-performing models still exhibit relatively high runtime error rates could be their tendency to pursue more aggressive and optimized coding patterns, such as employing custom data structures, in-place transformations, and pointer arithmetic. These advanced techniques, while algorithmically sound, might inherently increase the potential for execution faults[6], especially in edge scenarios. Interestingly, GPT-OSS-20B displays compilation error rates comparable to weaker, non-reasoning-intensive models. We attribute this unexpected result to its cautious approach: the model often declines to generate solutions when

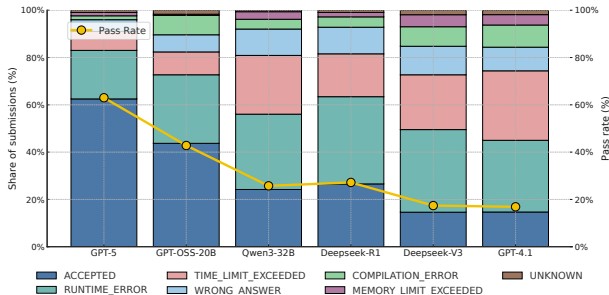

Figure 4: Submission status distribution for six selected models. The models are sorted based on performance from left to right. Solutions by stronger reasoning models show substantial reductions in failure types of time limit, memory limit, and compilation errors.

it anticipates insufficient reasoning time, thereby triggering compilation-related failures. These findings highlight a limitation in the reinforcement learning approaches employed by current models (DeepSeek-AI et al., 2025; Yang et al., 2025b), which predominantly use solution correctness as the sole reward, neglecting efficiency and memory management. Future training techniques could incorporate fine-grained reward signals targeting these attributes, enabling models to optimize not only for correctness but also for reliable and efficient code execution.

## 5.4 DATA CONTAMINATION STUDY

### 5.4.1 NO EVIDENCE OF TEMPORAL PERFORMANCE DEGRADATION.

In Figure A3 and Appendix F.1, we evaluate contamination risk by examining whether model performance correlates with problem release dates, particularly around each model's training cutoff. Our analysis shows no meaningful relationship between publication time and performance: models do not perform better on older tasks, nor do they exhibit any noticeable drop in accuracy on problems released after their knowledge cutoff. This lack of temporal correlation suggests that broad data leakage is unlikely and provides evidence that our benchmark is not driven by memorization effects tied to publication dates.

### 5.4.2 FAMILIARITY WITH TASK STATEMENTS AND OFFICIAL SOLUTIONS

Following MLE-Bench (Chan et al., 2024) on detecting data contamination, we investigate whether the models' familiarity with task statements and official solutions affects their performance. Intuitively, if a model is more familiar with a task, it tends to perform better. Specifically, we define a model's familiarity with a document as the mean probability it assigns to each token, conditioned on all preceding tokens. Using GPT-OSS-120B, we compute this familiarity metric and plotted it against various performance indicators, including pass rate (Figure A7) and relative score (Figure 5). Our analysis reveals a near-zero correlation between GPT-OSS-120B's familiarity with either task statements or official solutions and its performance. If a model's performance strongly correlated with higher familiarity scores, this would suggest potential memorization or overfitting of training data. However, our investigation demonstrates that familiarity with task statements does not significantly predict model performance, highlighting the minimal data contamination within our benchmark. If substantial template-based contamination exists within our benchmark, one would expect higher model familiarity with the official solutions to correlate strongly with improved performance, as the model would likely have encountered identical or similarly structured solutions during training.

---

[6]For instance, a simple algorithm like summing elements of an array becomes significantly more complex when highly optimized for memory access patterns using techniques such as loop unrolling and pragma directives in C++ (e.g., #pragma omp simd, #pragma unroll).

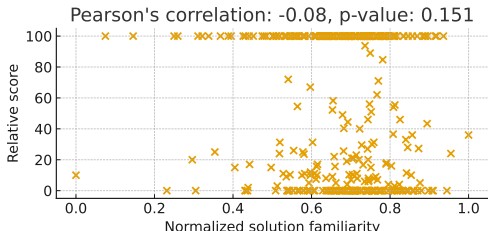 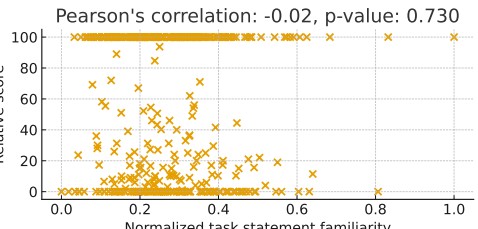

Figure 5: No significant positive correlation is observed between GPT-OSS-120B's familiarity with task statements and solutions (normalized via min-max scaling) and its performance, indicating that higher familiarity does not necessarily translate to better outcomes.

### 5.4.3 LOW SIMILARITY BETWEEN MODELS' SOLUTIONS AND OFFICIAL SOLUTIONS

Additionally, to investigate whether models rely primarily on memorization of template solutions rather than genuine reasoning, we employ the source code plagiarism detection tool Dolos (Maertens et al., 204) to measure the similarity between accepted solutions generated by GPT-5 and Grok-4-Fast-Reasoning and the official solutions. GPT-5 achieves a median similarity of 0.11, while Grok-4-Fast-Reasoning has a median similarity of 0.12. In Figure A8, the majority of these solutions exhibit similarity scores below 0.3, strongly indicating that the models do not simply reproduce memorized templates. Instead, the models appear to actively reason and generate novel solutions to solve each problem. In addition, we also find negligible correlation between template similarity and model performance differences, suggesting minimal template contamination (Appendix F.4 and Appendix F.7).

Based on these analyses, we believe that both direct data contamination and template-based contamination are minimized in our benchmark. Moreover, to continuously mitigate potential contamination risks, our benchmark will be regularly updated every 3-6 months with newly released official contests.

### 5.5 ADDITIONAL ANALYSES

Our Appendix provides additional analyses to complement the main results. Specifically, GPT-OSS-120B consistently performs best when generating solutions in C++, reflecting the language's inherent computational efficiency advantage (Appendix F.3). Further, our detailed examination of algorithmic tags indicates that increasing reasoning-token budgets yields greater improvements on complex tasks compared to parameter scaling or training strategy variations (Appendix F.5). Finally, we find minimal variance in model performance across different prompt variants, underscoring the model's robustness to instruction wording (Appendix F.6).

## 6 CONCLUSION

In this work, we propose LiveOIBench, a comprehensive competitive coding benchmark featuring expert-curated OI-style tasks with detailed subtask rubrics, direct comparisons to human contestant performance, continuous updates with new Olympiad tasks to prevent contamination, and an offline evaluation system ensuring accessible and reproducible assessments. We extensively evaluate 34 models including both proprietary and open-weight models. Our results highlight that proprietary models, particularly GPT-5, achieve impressive results but falls short of top human contestants, who typically place above the 90th percentile. Among open-source models, GPT-OSS, Seed-OSS, and Qwen-3-32B demonstrate significant progress, with GPT-OSS-120B notably narrowing the performance gap to proprietary alternatives. Further analyses reveal that current models particularly struggle with advanced algorithmic tasks, such as dynamic programming. Additionally, our reasoning trace analysis indicates that robust model performance relies on strategically allocating exploratory and analytical reasoning behaviors. Lastly, we find stronger models reduce common failures yet persistently face runtime errors due to optimized coding techniques, suggesting refined training for efficiency and memory management. Moving forward, we envision leveraging this benchmark to further investigate inference-time scaling strategies and training methods, particularly for challenging reasoning tasks. By offering a rigorous, reproducible, and continuously updated evaluation benchmark, LiveOIBench aims to drive significant advancements in the reasoning and coding capabilities of LLMs.

**Reproducibility Statement.** To ensure full reproducibility, we will publicly release all benchmark components. We provide detailed descriptions of our dataset construction process in Section 3 and Appendix B. Additionally, our evaluation code is openly accessible at `https://anonymous.4open.science/r/LiveOIBench-25F9/`.

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

## A  LLM Usage

In this work, we leverage LLMs to assist with refining and polishing the paper, as well as identifying relevant literature on code-generation benchmarks.

## B  Dataset Construction

### B.1  Task Collection

For each of the Informatics Olympiad Competitions identified, we collect all contests held after 2022 and record their official websites. We deliberately set the post-2022 timeframe in order to reduce the potential contamination from extensive pre-training datasets. The total number of contests retrieved is 72, with 46 of them containing human results data.

**Contest Information Extraction:** We then develop a web crawler for each contest and each year that can extract task information directly from official competition websites. The task information typically includes task statements, test cases, official and unofficial code solutions, code attachments, time and memory constraints, and subtask specifications for each problem. We also parse the contestant results page on each official website and reformat the results data into standardized CSV files. To accomplish this, we manually copy the raw webpage content into Gemini-2.0-Pro and prompt it to generate CSV files with normalized headers. These CSV files precisely capture the contestant names, countries, total scores, individual task scores, and awarded medals. We add them to our contestant database after carefully inspecting the file, making sure that it is consistent with the data from the website.

We also compute the threshold scores for Gold, Silver, and Bronze Medals, if applicable, by recording the lowest total score of all the participants that received the corresponding medal. For USACO Bronze, Silver, and Gold Contests, we use the minimum total score required to advance to the next level the threshold for the corresponding Bronze, Silver, or Gold medal. For the platinum contest, we determine medal thresholds based on the number of solved problems: participants solving exactly one problem earn a bronze medal, two solved problems qualify for silver, and more than two problems earns gold.

**Markdown Conversion Quality Check:** We sampled 40 tasks (around 10% of task statements) maintaining the same distribution across competition types as the full dataset. Our analysis revealed 4 tasks (10%) with conversion errors, primarily in nested table parsing within example inputs/outputs. While such formatting inconsistencies might impact human understanding, they have minimal effect on LLM comprehension as the core question semantic remains intact. GPT-5 fully solved three of the four tasks and passed over 70% of test cases on the remaining one, demonstrating that models are less sensitive to these formatting issues. An additional 14 tasks (35%) exhibited minor formatting issues, including header-level inconsistencies (e.g., "Input" formatted as  Input while "Output" is formatted as  Output despite being at the same hierarchical level), incorrect superscript rendering, and uppercase/lowercase mismatches. These are inconsequential formatting artifacts that do not alter the semantic meaning of the problems. Our conversion pipeline uses Marker[1] for PDF-to-markdown conversion followed by Gemini-2.0-Flash for automated verification and correction. After manually reviewing this 40-task sample, we confirmed that the parsed task statements are indeed of high quality, with only minor formatting errors that do not impact model performance. Confident in the robustness of these results, we proceeded with batch processing the remaining tasks. We believe these slight formatting inconsistencies are inconsequential and do not compromise the validity of our evaluation.

**Missing Data:** When the official website lacks complete or up-to-date contest information, we enhance our dataset by retrieving the missing details from reputable secondary platforms such as CSES and LibreOJ. These platforms host curated repositories of contest materials and metadata, and contain a substantial amount of user submissions along with their corresponding pass rates. Their widespread adoption within the competitive programming and informatics communities suggests high accuracy and reliability. For contests missing test cases on the official site, we employ a parser

to retrieve them from CSES and integrate them into our dataset. If official code solutions are absent or invalid, we obtain five user-submitted solutions from LibreOJ that achieved a $100\%$ pass rate and include them in the dataset. Valid solutions from open-source Github repositories are also downloaded to enhance the dataset. By supplementing incomplete primary data with these established sources, we ensure our dataset maintains high standards of accuracy and completeness.

## B.2    PROBLEM FILTERING AND SOLUTION VERIFICATION

To ensure that the solutions collected from official websites and external platforms are accurate, we create an evaluation code judge to testify whether our solution can pass all the test cases from our dataset. The code judge operates differently based on the question type.

**Batch:**  For all the batch problems, we run the official code solution against the input-output test cases. The input file is provided to the program, and the code output is verified against the expected output. The subtask scores are computed to verify that the total score adds up to 100 points. Any problem for which the solution failed a test case or produced an invalid total score was excluded from further analysis. For problems that accept multiple valid outputs, we set up a testing environment using the grader file supplied in the contest materials and apply the same evaluation procedure, disregarding problems with incorrect solution files.

**Interactive:**  If the problem type is interactive, the grader file is executed first to establish the testing environment. Subsequently, the solution file is launched within the environment to exchange input/output streams interactively. After the problem finishes, the grader's evaluation output is collected to determine whether the solution passed. If the grader doesn't return a full mark on the ground-truth solution, the corresponding problem is discarded.

**Output-Only:**  We exclude output only problems since they don't require algorithmic code solutions, making them unsuitable for evaluating model performance.

## B.3    METADATA COLLECTION

To further enrich and structure our dataset, we augment it with comprehensive problem metadata crawled from solved.ac and Luogu , capturing difficulty ratings and algorithm tags. We then utilize Gemini-2.0-Flash to semantically match problems across different platforms, resolving inconsistencies in label formats and taxonomies through a unified mapping strategy.

**Difficulty Tags:**  Solved.ac uses integer values from 1 to 30 to represent the difficulty levels, where 1 corresponds to the easiest tier (Bronze V) and 30 corresponds to the hardest tier (Ruby I). In contrast, Luogu categorizes problem difficulty into 7 textual levels. To align the two difficulty systems, we construct a numerical mapping on a $0 - 30$ scale for Luogu, translating the native difficulty descriptor tags into standardized numerical scores using the mapping as specified in Table A1. This normalization provides a consistent basis for assigning comparable difficulty scores to the problems.

| Difficulty Tag | Difficulty Score |
| --- | --- |
| Beginner | 5 |
| Easy | 9 |
| Intermediate | 13 |
| Hard | 16 |
| Advanced | 18 |
| Expert | 21 |
| Master | 24 |

Table A1: Difficulty Tags and Corresponding Scores

**Algorithm Tags.** To ensure data integrity and consistency, we develop a normalization dictionary to standardize dataset labels. This dictionary systematically resolves lexical and semantic variations, including synonyms, related terms, and differences in granularity, by mapping them to a unified set of canonical tags.

**Missing Tags.** In cases where algorithm tags were missing or where difficulty tags were by default at 0, we utilize Gemini-2.0-Flash to infer plausible algorithm and difficulty labels from the problem description, enhancing both completeness and labeling quality. To assess the reliability of LLM-

inferred difficulty scores, we conduct sampling-based validation on problems with existing difficulty annotations and observe a high degree of consistency with their original scores.

**Divisions.** Finally, we analyze the distribution of algorithm and difficulty tags across the corpus and partition the difficulty range of all contests into four divisions, thereby improving robustness and facilitating downstream contest categorization. The division boundaries are listed in Table A2.

| Division | Min Difficulty | Max Difficulty | Avg Difficulty | Total Contests |
|---|---|---|---|---|
| Division 4 | 5.0 | 15.78 | 13.76 | 17 |
| Division 3 | 16.0 | 20.33 | 18.05 | 19 |
| Division 2 | 20.33 | 22.33 | 21.52 | 19 |
| Division 1 | 22.5 | 30.0 | 23.62 | 17 |

Table A2: Division Boundaries by Difficulty

**Problem Difficulty.** We sort all task difficulty scores and split them into three equal-sized buckets by taking the empirical one-third and two-thirds cut points. The problem difficulty distribution is listed in Table A3

Table A3: Problem difficulty distribution using quantile thresholds.

| Level | # Problems | % of Total | Threshold Rule |
|---|---|---|---|
| Easy | 143 | 35.48% | $d \leq 17$ |
| Medium | 144 | 35.73% | $18 \leq d \leq 22$ |
| Hard | 116 | 28.78% | $d \geq 23$ |
| **Total** | **403** | **100%** | – |

### B.4 CODEFORCES RATINGS COLLECTION

**Result Collection:** For each contest, the raw results files are downloaded and restructured. These files typically include contestant identifiers such as usernames, countries, individual task scores, total scores, and medal information.

**Rating Data Retrieval:** Codeforces rating data are obtained by algorithmically mapping contestants' names to their corresponding profiles in the Codeforces database. Usernames are first normalized by removing diacritics and converting all text to lowercase to enhance matching robustness. Using these normalized usernames together with the contestant's country, the system issues Google Search queries and evaluates the top results for potential Codeforces profile URLs. When valid Codeforces URLs are identified, the extracted handles are queried via the Codeforces API to obtain detailed user profile information, including full name, country, Codeforces ID, and rating history.

**Database Generation:** The retrieved rating histories are parsed to extract Codeforces ratings for each year from 2022 to 2025. Where annual data are missing, a backfilling procedure incorporates information from prior years, where applicable. Contest names and contestant metadata are then appended to a master Codeforces database. If a contestant's record already exists, only new contest information is added to the existing profile.

**Rating Matching:** After these procedures are applied to every contest results file, a comprehensive database of Codeforces ratings for all contestants is established. Finally, each contest record is linked to the Codeforces rating corresponding to the year of that contest.

**Model Ratings:** To benchmark the performance of different models against our dataset, we assign each model Codeforces ratings for each contest. For each task, we present the full problem statement to the models and prompted them to generate code solutions. Using the provided subtask and test-case data, we compute the total score of each model's solution. Once total scores were obtained, we derive Codeforces ratings for the models using the CodeElo formula (Quan et al., 2025) given below, aligning model performance with that of human participants.

$$m = \sum_{i=1}^{n} \frac{1}{1 + 10^{\frac{r - r_{(i)}}{400}}}$$

To ensure data quality, we apply several preprocessing steps to the human results before computing ELO scores. We exclude all participants who lacked an official Codeforces rating or whose rating was below $500$, ensuring that only valid ratings were used. We further remove outliers by fitting a third-degree polynomial regression between total score and Codeforces rating, removing participants whose performance deviates by more than 2 standard deviations from the regression line.

Finally, to mitigate noise and improve reliability, we exclude contests with fewer than $15$ valid human Codeforces ratings from our Elo calculations. These procedures ensure that the resulting ratings accurately reflect the relationship between Codeforces ratings and total score for human participants.

**Codeforces Profile Quality Check:**    To verify the accuracy of matching contestants to their Codeforces profiles, we performed a detailed manual error analysis. We randomly sampled five contest results, selecting ten contestants from each contest. Using the same search query (contestant name, country, and "Codeforces"), we located each contestant's Codeforces profile and verified the match by comparing the name, country, username, and yearly rating history against our stored records.. Our analysis confirmed that all 50 contestant profiles matched perfectly, demonstrating 100% accuracy and confirming the high precision of our profile-matching pipeline. Nevertheless, we recognize that some contestants may lack Codeforces profiles or use alternative names on their profiles, potentially causing missed matches. Given the high precision of our pipeline, the individualized nature of ELO rating computations, and our exclusion of contests containing fewer than 15 validated contestant profiles, we anticipate that such omissions will minimally affect the overall accuracy of the model's rating estimations.

Table A4: Dataset Statistics across Various Programming Competitions

| Competitions | Total Tasks | Total Contests | Avg. Subtasks | Test Cases/Task | Token Count | Difficulty |
|---|---|---|---|---|---|---|
| IOI | 12 | 2 | 7.08 | 112.42 | 2359.58 | 22.83 |
| BOI | 18 | 3 | 6.22 | 110.83 | 1139.72 | 22.28 |
| CEOI | 11 | 2 | 7.45 | 89.45 | 1339.36 | 22.33 |
| EGOI | 13 | 2 | 5.31 | 87.23 | 1388.85 | 18.50 |
| EJOI | 12 | 2 | 7.25 | 54.92 | 1443.08 | 12.00 |
| IATI | 11 | 2 | 6.82 | 78.09 | 1302.00 | 23.03 |
| OOI | 32 | 4 | 8.88 | 128.19 | 1639.31 | 23.02 |
| RMI | 12 | 2 | 6.33 | 37.42 | 896.42 | 23.00 |
| APIO | 5 | 2 | 8.00 | 58.80 | 2052.40 | 21.67 |
| JOI | 42 | 7 | 5.79 | 103.00 | 1848.29 | 21.17 |
| CCO | 32 | 6 | 4.19 | 63.34 | 754.25 | 13.36 |
| COCI | 62 | 13 | 3.69 | 55.02 | 897.05 | 16.38 |
| NOI | 9 | 3 | 6.11 | 63.22 | 970.00 | 21.89 |
| USACO | 132 | 22 | - | 17.11 | 751.07 | 19.13 |
| Division 1 | 87 | 17 | 7.42 | 85.45 | 1440.46 | 23.62 |
| Division 2 | 89 | 19 | 5.23 | 80.16 | 1288.83 | 21.52 |
| Division 3 | 115 | 19 | 7.32 | 57.85 | 1124.07 | 18.05 |
| Division 4 | 112 | 17 | 3.80 | 28.54 | 738.32 | 13.76 |
| All Competitions | 403 | 72 | 5.80 | 60.59 | 1121.55 | 19.04 |

Table A5: Contest dates from 2023–2025 for major Olympiads.

| Contest | Date | Human Results |
|---|---|---|
| Asia-Pacific Informatics Olympiad 2023 | 2023-05-20 | True |
| Asia-Pacific Informatics Olympiad 2024 | 2024-05-18 | True |
| Baltic Olympiad in Informatics 2023 | 2023-04-28 | True |
| Baltic Olympiad in Informatics 2024 | 2024-05-03 | True |
| Baltic Olympiad in Informatics 2025 | 2025-04-29 | True |
| Canadian Computing Olympiad 2023 CCC_Junior | 2023-02-15 | True |
| Canadian Computing Olympiad 2023 CCC_Senior | 2023-02-15 | True |
| Canadian Computing Olympiad 2023 CCO | 2023-05-29 | True |
| Canadian Computing Olympiad 2024 CCC_Junior | 2024-02-21 | True |
| Canadian Computing Olympiad 2024 CCC_Senior | 2024-02-27 | True |
| Canadian Computing Olympiad 2024 CCO | 2024-05-27 | False |
| Central European Olympiad in Informatics 2023 | 2023-08-13 | True |
| Central European Olympiad in Informatics 2024 | 2024-06-24 | True |
| Croatian Open Competition in Informatics 2023 CONTEST_#3 | 2023-01-14 | True |
| Croatian Open Competition in Informatics 2023 CONTEST_#4 | 2023-02-11 | True |
| Croatian Open Competition in Informatics 2023 CONTEST_#5 | 2023-03-11 | True |
| Croatian Open Competition in Informatics 2024 CONTEST_#1 | 2023-11-04 | True |
| Croatian Open Competition in Informatics 2024 CONTEST_#2 | 2023-12-02 | True |
| Croatian Open Competition in Informatics 2024 CONTEST_#3 | 2024-01-13 | True |
| Croatian Open Competition in Informatics 2024 CONTEST_#4 | 2024-02-10 | True |
| Croatian Open Competition in Informatics 2024 CONTEST_#5 | 2024-03-16 | True |
| Croatian Open Competition in Informatics 2025 CONTEST_#1 | 2024-10-05 | True |
| Croatian Open Competition in Informatics 2025 CONTEST_#2 | 2024-11-09 | True |
| Croatian Open Competition in Informatics 2025 CONTEST_#3 | 2024-12-07 | True |
| Croatian Open Competition in Informatics 2025 CONTEST_#4 | 2025-01-25 | True |
| Croatian Open Competition in Informatics 2025 CONTEST_#5 | 2025-02-15 | True |
| European Girls' Olympiad in Informatics 2023 | 2023-07-15 | True |
| European Girls' Olympiad in Informatics 2024 | 2024-07-21 | True |
| European Junior Olympiad in Informatics 2023 | 2023-09-08 | True |
| European Junior Olympiad in Informatics 2024 | 2024-08-16 | True |
| International Advanced Tournament in Informatics 2024 junior | 2024-04-17 | False |
| International Advanced Tournament in Informatics 2024 senior | 2024-04-17 | False |
| International Olympiad in Informatics 2023 | 2023-08-28 | True |
| International Olympiad in Informatics 2024 | 2024-09-01 | True |
| Japanese Olympiad in Informatics 2023 JOI | 2023-02-12 | True |
| Japanese Olympiad in Informatics 2023 JOI_open | 2023-08-05 | True |
| Japanese Olympiad in Informatics 2023 JOI_spring | 2023-03-19 | True |
| Japanese Olympiad in Informatics 2024 JOI | 2024-02-04 | True |
| Japanese Olympiad in Informatics 2024 JOI_open | 2024-06-17 | True |
| Japanese Olympiad in Informatics 2024 JOI_spring | 2024-03-21 | True |
| Japanese Olympiad in Informatics 2025 JOI | 2025-02-02 | True |
| Nordic Olympiad in Informatics 2023 | 2023-03-22 | True |
| Nordic Olympiad in Informatics 2024 | 2024-03-06 | True |
| Nordic Olympiad in Informatics 2025 | 2025-03-05 | True |
| Open Olympiad in Informatics 2023 final | 2024-03-07 | True |
| Open Olympiad in Informatics 2023 qualification | 2023-11-25 | True |
| Open Olympiad in Informatics 2024 final | 2025-03-06 | True |
| Open Olympiad in Informatics 2024 qualification | 2024-12-01 | True |
| Romanian Master of Informatics 2023 | 2023-10-11 | True |
| Romanian Master of Informatics 2024 | 2024-11-27 | True |
| USA Computing Olympiad 2023 December_Contest-combined | 2022-12-15 | False |
| USA Computing Olympiad 2023 December_Contest-platinum | 2022-12-15 | False |

| Contest | Date | Human Results |
|---|---|---|
| USA Computing Olympiad 2023 February_Contest-combined | 2023-02-24 | False |
| USA Computing Olympiad 2023 February_Contest-platinum | 2023-02-24 | False |
| USA Computing Olympiad 2023 January_Contest-combined | 2023-01-27 | False |
| USA Computing Olympiad 2023 January_Contest-platinum | 2023-01-27 | False |
| USA Computing Olympiad 2023 US_Open_Contest-combined | 2023-03-24 | False |
| USA Computing Olympiad 2023 US_Open_Contest-platinum | 2023-03-24 | False |
| USA Computing Olympiad 2024 December_Contest-combined | 2023-12-13 | False |
| USA Computing Olympiad 2024 December_Contest-platinum | 2023-12-13 | False |
| USA Computing Olympiad 2024 February_Contest-combined | 2024-02-16 | False |
| USA Computing Olympiad 2024 February_Contest-platinum | 2024-02-16 | False |
| USA Computing Olympiad 2024 January_Contest-combined | 2024-01-26 | False |
| USA Computing Olympiad 2024 January_Contest-platinum | 2024-01-26 | False |
| USA Computing Olympiad 2024 US_Open_Contest-combined | 2024-03-15 | False |
| USA Computing Olympiad 2024 US_Open_Contest-platinum | 2024-03-15 | False |
| USA Computing Olympiad 2025 February_Contest-combined | 2025-02-21 | False |
| USA Computing Olympiad 2025 February_Contest-platinum | 2025-02-21 | False |
| USA Computing Olympiad 2025 January_Contest-combined | 2025-01-24 | False |
| USA Computing Olympiad 2025 January_Contest-platinum | 2025-01-24 | False |
| USA Computing Olympiad 2025 US_Open_Contest-combined | 2025-03-21 | False |
| USA Computing Olympiad 2025 US_Open_Contest-platinum | 2025-03-21 | False |
| Total: 72 | | 46 |

Table A6: Summary of Human Codeforces ratings for various contests.

| Contest | Contestants | Average Rating |
|---|---|---|
| Asia-Pacific Informatics Olympiad 2023 | 60 | 2184.85 |
| Asia-Pacific Informatics Olympiad 2024 | 72 | 2108.28 |
| Baltic Olympiad in Informatics 2023 | 24 | 2006.12 |
| Baltic Olympiad in Informatics 2024 | 27 | 1973.11 |
| Baltic Olympiad in Informatics 2025 | 19 | 2023.37 |
| Canadian Computing Olympiad 2023 CCC_Junior | 185 | 1993.04 |
| Canadian Computing Olympiad 2023 CCC_Senior | 88 | 2141.22 |
| Canadian Computing Olympiad 2023 CCO | 7 | 2379.14 |
| Canadian Computing Olympiad 2024 CCC_Junior | 228 | 1822.74 |
| Canadian Computing Olympiad 2024 CCC_Senior | 98 | 1960.28 |
| Central European Olympiad in Informatics 2023 | 28 | 2214.57 |
| Central European Olympiad in Informatics 2024 | 27 | 2156.81 |
| Croatian Open Competition in Informatics 2023 CONTEST_#3 | 10 | 2050.7 |
| Croatian Open Competition in Informatics 2023 CONTEST_#4 | 10 | 2050.7 |
| Croatian Open Competition in Informatics 2023 CONTEST_#5 | 10 | 2050.7 |
| Croatian Open Competition in Informatics 2024 CONTEST_#1 | 65 | 1795.92 |
| Croatian Open Competition in Informatics 2024 CONTEST_#2 | 55 | 1807.35 |
| Croatian Open Competition in Informatics 2024 CONTEST_#3 | 61 | 1873.16 |
| Croatian Open Competition in Informatics 2024 CONTEST_#4 | 55 | 1756.38 |
| Croatian Open Competition in Informatics 2024 CONTEST_#5 | 58 | 1744.55 |
| Croatian Open Competition in Informatics 2025 CONTEST_#1 | 5 | 2016.6 |
| Croatian Open Competition in Informatics 2025 CONTEST_#2 | 5 | 2016.6 |
| Croatian Open Competition in Informatics 2025 CONTEST_#3 | 5 | 2016.6 |
| Croatian Open Competition in Informatics 2025 CONTEST_#4 | 5 | 2016.6 |
| Croatian Open Competition in Informatics 2025 CONTEST_#5 | 5 | 2016.6 |
| European Girls' Olympiad in Informatics 2023 | 54 | 1646.02 |
| European Girls' Olympiad in Informatics 2024 | 31 | 1678.23 |
| European Junior Olympiad in Informatics 2023 | 22 | 1876.0 |
| European Junior Olympiad in Informatics 2024 | 32 | 1877.16 |
| International Olympiad in Informatics 2023 | 216 | 2105.12 |
| International Olympiad in Informatics 2024 | 253 | 2115.76 |
| Japanese Olympiad in Informatics 2023 JOI | 139 | 2314.65 |
| Japanese Olympiad in Informatics 2023 JOI_open | 98 | 2195.65 |
| Japanese Olympiad in Informatics 2023 JOI_spring | 252 | 2278.29 |
| Japanese Olympiad in Informatics 2024 JOI | 144 | 2022.38 |
| Japanese Olympiad in Informatics 2024 JOI_open | 102 | 2263.97 |
| Japanese Olympiad in Informatics 2024 JOI_spring | 245 | 2221.79 |
| Nordic Olympiad in Informatics 2023 | 16 | 1695.5 |
| Nordic Olympiad in Informatics 2024 | 13 | 1726.08 |
| Nordic Olympiad in Informatics 2025 | 6 | 1687.67 |
| Open Olympiad in Informatics 2023 final | 142 | 2028.51 |
| Open Olympiad in Informatics 2023 qualification | 92 | 1421.75 |
| Open Olympiad in Informatics 2024 final | 69 | 2037.86 |
| Open Olympiad in Informatics 2024 qualification | 87 | 1512.4 |
| Romanian Master of Informatics 2023 | 75 | 1953.19 |
| Romanian Master of Informatics 2024 | 93 | 1970.59 |

## B.5   SAMPLE TASK

We now present an example drawn from the *International Olympiad in Informatics 2024*. The following task, titled *Nile*, illustrates a typical problem style in our dataset.

**Problem: Nile**

You want to transport $N$ artifacts through the Nile. The artifacts are numbered from 0 to $N-1$. The weight of artifact $i$ ($0 \leq i < N$) is $W[i]$.

To transport the artifacts, you use specialized boats. Each boat can carry **at most two** artifacts.

- If you decide to put a single artifact in a boat, the artifact weight can be arbitrary.
- If you want to put two artifacts in the same boat, you have to make sure the boat is balanced evenly. Specifically, you can send artifacts $p$ and $q$ ($0 \leq p < q < N$) in the same boat only if the absolute difference between their weights is at most $D$, i.e. $|W[p] - W[q]| \leq D$.

The cost of transporting artifact $i$ ($0 \leq i < N$) is:

- $A[i]$, if you put the artifact in its own boat, or
- $B[i]$, if you put it in a boat together with some other artifact.

If artifacts $p$ and $q$ are sent together, the total cost is $B[p] + B[q]$. Since $B[i] < A[i]$ for all $i$, sending an artifact with another is always cheaper when possible.

Unfortunately, the river is unpredictable and the value of $D$ changes often. Your task is to answer $Q$ queries, described by array $E$ of length $Q$. For query $j$ ($0 \leq j < Q$), the answer is the minimum cost of transporting all $N$ artifacts when $D = E[j]$.

**Implementation Details**

```
std::vector<long long> calculate_costs(
    std::vector<int> W,
    std::vector<int> A,
    std::vector<int> B,
    std::vector<int> E)
```

- `W`, `A`, `B`: arrays of length $N$, describing weights and costs.
- `E`: array of length $Q$, values of $D$.
- Returns: array $R$ with $R[j]$ equal to the minimum cost for $D = E[j]$.

**Constraints**

$1 \leq N \leq 100,000$
$1 \leq Q \leq 100,000$
$1 \leq W[i] \leq 10^9$ for each $i$ such that $0 \leq i < N$
$1 \leq B[i] < A[i] \leq 10^9$ for each $i$ such that $0 \leq i < N$
$1 \leq E[j] \leq 10^9$ for each $j$ such that $0 \leq j < Q$

**Subtasks**

| Subtask | Score | Additional Constraints |
|---------|-------|------------------------|
| 1 | 6 | $Q \leq 5$; $N \leq 2000$; $W[i] = 1$ for each $i$ such that $0 \leq i < N$ |
| 2 | 13 | $Q \leq 5$; $W[i] = i + 1$ for each $i$ such that $0 \leq i < N$ |
| 3 | 17 | $Q \leq 5$; $A[i] = 2$ and $B[i] = 1$ for each $i$ such that $0 \leq i < N$ |
| 4 | 11 | $Q \leq 5$; $N \leq 2000$ |
| 5 | 20 | $Q \leq 5$ |
| 6 | 15 | $A[i] = 2$ and $B[i] = 1$ for each $i$ such that $0 \leq i < N$ |
| 7 | 18 | No additional constraints. |

**Example**

```
calculate_costs([15, 12, 2, 10, 21],
                [5, 4, 5, 6, 3],
                [1, 2, 2, 3, 2],
                [5, 9, 1]) -> [16, 11, 23]
```

Explanation:

- $D = 5$: pair $(0, 3)$, others alone $\Rightarrow$ 16
- $D = 9$: pairs $(0, 1)$ and $(2, 3)$, artifact 4 alone $\Rightarrow$ 11

- $D = 1$: no pairs possible, all alone $\Rightarrow 23$

**Sample Grader**

*Input format:*

```
N
W[0] A[0] B[0]
W[1] A[1] B[1]
...
W[N-1] A[N-1] B[N-1]
Q
E[0]
E[1]
...
E[Q-1]
```

*Output format:*

```
R[0]
R[1]
...
R[S-1]
```

where $S = Q$ is the length of the output array.

### grader.cpp

```cpp
#include "nile.h"
#include <cstdio>
#include <vector>

int main() {
  int N; scanf("%d", &N);
  std::vector<int> W(N), A(N), B(N);
  for (int i = 0; i < N; i++)
    scanf("%d%d%d", &W[i], &A[i], &B[i]);
  int Q; scanf("%d", &Q);
  std::vector<int> E(Q);
  for (int j = 0; j < Q; j++)
    scanf("%d", &E[j]);

  auto R = calculate_costs(W, A, B, E);
  for (auto x : R) printf("%lld\n", x);
}
```

### Problem Metadata

```
"nile": {
        "id": 32262,
        "title": "Pyramids",
        "difficulty": 7,
        "tags": ["prefix sum"],
        "time_limit": 2.0,
        "memory_limit": 2048.0,
        "task_type": "Batch"
      }
```

## B.6 COMPETITION INFORMATION

**International Olympiad in Informatics (IOI)**   First held in 1989, the IOI is the annual world championship for informatics. Participants are organized into national delegations, with each of the approximately 90 participating countries sending a team of up to four students. These contestants are selected through highly rigorous, multi-stage national olympiads.

**Baltic Olympiad in Informatics (BOI)**   Established in 1995, the BOI brings together teams from countries bordering the Baltic Sea and invited guest nations. Each member country's national informatics organization selects a team of their top-ranking secondary school students, who are often candidates for that year's IOI team.

**Central European Olympiad in Informatics (CEOI)**   Originating in 1994, the CEOI is an on-site competition for teams from Central European member countries and several guest nations. Delegations are chosen by respective national olympiad committees and are typically composed of students who have achieved top results in their national contests.

**European Girls' Olympiad in Informatics (EGOI)**   An initiative from 2021, the EGOI is an inter-national competition for teams from European and guest countries. Each participating country selects a team of up to four female secondary school students who have demonstrated strong performance in their national-level informatics competitions.

**European Junior Olympiad in Informatics (EJOI)**   Founded in 2017, the EJOI is a major international event for a younger age group. Each European member country sends a national delegation of up to four students who are under the age of 15.5. Participants are typically the winners of national junior-level informatics olympiads.

**International Advanced Tournament in Informatics (IATI)**   Established in 2009 and hosted in Shumen, Bulgaria, the IATI is an international competition with two distinct age divisions, Junior and Senior. It brings together national and regional teams from numerous participating countries. Contestants are typically selected by their national informatics organizations based on strong results in previous competitions.

**Open Olympiad in Informatics (OOI)**   The Open Olympiad in Informatics (OOI) is the final stage of the All-Russian Olympiad in Informatics. Its participants are composed of two groups: the top Russian students who have advanced through a rigorous nationwide selection process, and official teams from various guest countries that receive a formal invitation to compete.

**Romanian Master of Informatics (RMI)**   First held in 2009, the RMI is a prestigious international competition. Participation is by invitation only; the organizers invite official national teams from countries with a strong track record at the IOI. This makes the participant pool one of the strongest in the world.

**Asia-Pacific Informatics Olympiad (APIO)**   The APIO, an online contest since 2007, involves students from countries and regions across the Asia-Pacific. Each member region organizes its own contest to select a set of national participants, who then compete from a supervised site within their home country.

**Japanese Olympiad in Informatics (JOI)**   Since 1994, the JOI has served as Japan's national selection process. It is open to Japanese junior high and high school students, who compete in preliminary rounds. Top performers are then invited to an exclusive on-site final and training camp, from which the IOI team is chosen.

**Canadian Computing Olympiad (CCO)**   The CCO, since 1996, is the invitational final stage of Canada's national selection process. Participation is granted to the top $20 - 25$ senior-level students from the open Canadian Computing Competition (CCC), who then compete to form the four-member IOI team.

**Croatian Open Competition in Informatics (COCI)**   Since 2006, COCI has operated as an online contest series open to individual participants worldwide. For Croatian students, cumulative performance across the year's rounds is a primary component in the selection process for the national team for the IOI and other international events.

**Nordic Olympiad in Informatics (NOI)**   The Nordic Olympiad in Informatics brings together top secondary school students from Denmark, Finland, Iceland, Norway, and Sweden. Each country selects its participants based on the results of their respective national olympiads, with the NOI serving as a key qualifier for the BOI.

**USA Computing Olympiad (USACO)**   The USACO is an open competition primarily for pre-college students in the United States, though it attracts many international participants. Its monthly online contests determine which top US-based students in the Platinum division are invited to a training camp, where the four-member IOI team is selected.

## C   MODEL INFORMATION

- **Proprietary LLMs:** This category includes high-performing proprietary models such as Gemini-2.5 (Comanici et al., 2025), GPT-O3-Mini-High (OpenAI, 2025c), and GPT-4.1 (OpenAI, 2025a).
- **Open-Source Thinking LLMs:** These are openly available models that are empowered with inherent thinking or reasoning capabilities. This group includes Qwen3 (Yang et al., 2025a) and DeepSeek-R1 (DeepSeek-AI et al., 2025), as well as those distlled from DeepSeek-R1.
- **Open-Source Non-Thinking LLMs:** This category consists of openly available models that are not equipped with instinsic thinking mechanisms. This includes DeepSeek Coder-V2 (DeepSeek-AI et al., 2024b), DeepSeek-V3 (DeepSeek-AI et al., 2024a), Qwen2.5 (Yang et al., 2024), Qwen2.5-Coder (Hui et al., 2024), Qwen3 (Yang et al., 2025a), Mistral (Jiang et al., 2023) and LLaMa-3 (Dubey et al., 2024).
- Refer to Table A7 and Table A8 for more details. We use the default parameter setting if it's not listed.
- We use the prompt at Table C for all models.

Table A7: Model list of Non-Thinking LLMs with model providers and decoding settings

| Non-Thinking LLMs | Model Provider | Temperature | Top-p |
|---|---|---|---|
| GPT-4.1 (OpenAI, 2025a) | OpenAI | 1.0 | 1.0 |
| Qwen2.5-72B-Instruct (Yang et al., 2024) | Alibaba | 0.7 | 0.8 |
| Qwen2.5-Coder-32B-Instruct (Hui et al., 2024) | Alibaba | 0.7 | 0.8 |
| Qwen2.5-Coder-14B-Instruct (Hui et al., 2024) | Alibaba | 0.7 | 0.8 |
| Qwen2.5-Coder-7B-Instruct (Hui et al., 2024) | Alibaba | 0.7 | 0.8 |
| Mistral-Large-Instruct-2411 (Jiang et al., 2023) | Mistral | 1.0 | 1.0 |
| Mistral-Small-3.1-24B-2503 (Jiang et al., 2023) | Mistral | 1.0 | 1.0 |
| LLaMa-4-Scout (Meta AI, 2025) | Meta | 0.6 | 0.9 |
| LLaMa-3.3-70B-Instruct (Dubey et al., 2024) | Meta | 0.6 | 0.9 |
| LLaMa-3.1-8B-Instruct (Dubey et al., 2024) | Meta | 0.6 | 0.9 |
| DeepSeek-V3 (DeepSeek-AI et al., 2024a) | DeepSeek | 1.0 | 1.0 |
| DeepSeek-Coder-V2-Lite-Instruct (DeepSeek-AI et al., 2024b) | DeepSeek | 0.3 | 0.95 |
| Codestral-22B-v0.1 (Mistral AI team, 2024) | Mistral | 0.0 | 1.0 |

Table A8: Model list with categories, including model names, organizations, reasoning budget, and decoding settings.

| Thinking LLMs | Model Provider | Reasoning Budget (Max Tokens) | Temperature | Top-p |
|---|---|---|---|---|
| GPT-5 (OpenAI, 2025b) | OpenAI | Medium (100K) | - | - |
| GPT-O3-Mini-High (OpenAI, 2025c) | OpenAI | High (100K) | - | - |
| GPT-OSS-120B-High (OpenAI et al., 2025) | OpenAI | High (128K) | 1.0 | 1.0 |
| GPT-OSS-20B-High (OpenAI et al., 2025) | OpenAI | High (128K) | 1.0 | 1.0 |
| GPT-OSS-120B (OpenAI et al., 2025) | OpenAI | Medium (128K) | 1.0 | 1.0 |
| GPT-OSS-20B (OpenAI et al., 2025) | OpenAI | Medium (128K) | 1.0 | 1.0 |
| Grok-4-Fast-Reasoning (xAI, 2025) | xAI | 100K | 1.0 | 1.0 |
| Claude-Sonnet-4.5 (Anthropic, 2025) | Anthropic | 120K | 1.0 | 1.0 |
| SEED-OSS (ByteDance Seed Team, 2025) | ByteDance | Unlimited (128K) | 1.1 | 0.95 |
| Qwen3-32B (Yang et al., 2025a) | Alibaba | 38K | 0.6 | 0.95 |
| Qwen3-14B (Yang et al., 2025a) | Alibaba | 38k | 0.6 | 0.95 |
| QwQ-32B (Qwen Team, 2025) | Alibaba | 32K | 0.6 | 0.95 |
| Qwen3-30B (Yang et al., 2025a) | Alibaba | 38K | 0.6 | 0.95 |
| Qwen3-8B (Yang et al., 2025a) | Alibaba | 38K | 0.6 | 0.95 |
| Qwen3-4B (Yang et al., 2025a) | Alibaba | 38k | 0.6 | 0.95 |
| Gemini-2.5-Pro-exp-03-25 (Comanici et al., 2025) | Google | 64K | 1.0 | 0.95 |
| Gemini-2.5-Flash-preview-04-17 (Comanici et al., 2025) | Google | 64K | 1.0 | 0.95 |
| DeepSeek-R1-01-28 (DeepSeek-AI et al., 2025) | DeepSeek | 32K | 0.6 | 0.95 |
| DeepSeek-R1-Distill-Llama-70B (DeepSeek-AI et al., 2025) | DeepSeek | 32K | 0.6 | 0.95 |
| DeepSeek-R1-Distill-Qwen-32B (DeepSeek-AI et al., 2025) | DeepSeek | 32K | 0.6 | 0.95 |
| DeepSeek-R1-Distill-Qwen-14B (DeepSeek-AI et al., 2025) | DeepSeek | 32K | 0.6 | 0.95 |
| DeepSeek-R1-Distill-Llama-8B (DeepSeek-AI et al., 2025) | DeepSeek | 32K | 0.6 | 0.95 |

---

### Prompt Setup

Below are the instructions we use to evaluate the models, and we also include this prompt template in our appendix.

Given a competition problem below, write a solution in C++ that solves all the subtasks. Make sure to wrap your code in '""<task>.cpp' and '"''' Markdown delimiters.

[BEGIN PROBLEM]
<task statement>
[END PROBLEM]

Time limit: <time> seconds
Memory limit: <Memory> MB

Generate a solution in C++ that solves the task. Make sure to wrap your code in '""<task>.cpp' and '"''' Markdown delimiters.

- - - - - - - - - - - - - - - - - - - - - - - - - - - - - - - - - - - - - - - - -

Prompt configuration used in all experiments.

## D    EVALUATION METRICS

- **Pass@k (Kulal et al., 2019; Chen et al., 2021b):** We use the conventional Pass@k, which measures the fraction of problems for which at least one of the $k$ generated solutions is correct. We use $k = 8$.

- **Relative Score:** This metric is defined as the division of the model's score over the total possible score of a contest, providing a normalized measure of performance.

- **Average Percentile:** To benchmark LLM performance against human capabilities, we map the models' scores to a percentile rank based on the performance distribution of human contestants.

- **Olympics Medal System:** It uses the authoritative cutoffs in the Olympiads to decide if a model's performance is qualified for a medal (gold, silver, or bronze).

- **Codeforces ELO:** Inspired by the widely used rating system in competitive programming, we treat each model as a "virtual contestant" and update its rating after every contest based on its relative standing against human participants.

# E  FULL RESULTS

We present the complete evaluation results of all models on LiveOIBench. Table A9 provides the overall leaderboard across all 72 contests, while Table A10 breaks down performance by contest tags. Finally, Figure A1 shows a screenshot of the LiveOIBench website, which allows users to interactively explore model performances by selecting specific contest ranges.

| Model | Medals (%) | | | | Relative Score (%) | Human Percentile (%) | | | | | Pass Rate (%) | | | | CF Rating | | | | |
|---|---|---|---|---|---|---|---|---|---|---|---|---|---|---|---|---|---|---|---|
| | Gold | Silver | Bronze | Total | | All | D1 | D2 | D3 | D4 | All | Easy | Med. | Hard | All | D1 | D2 | D3 | D4 |
| **Proprietary LLMs** | | | | | | | | | | | | | | | | | | | |
| GPT-5 | 50.00 | 30.56 | 8.33 | 88.89 | 67.21 | 81.76 | 73.68 | 79.30 | 84.87 | 97.20 | 63.03 | 92.25 | 60.42 | 34.19 | 2414 | 2426 | 2322 | 2412 | 2583 |
| Grok-4-Fast-Reasoning | 45.83 | 20.83 | 16.67 | 83.33 | 56.99 | 74.23 | 63.79 | 67.62 | 79.14 | 97.46 | 50.95 | 70.42 | 39.58 | 15.38 | 2221 | 2053 | 2158 | 2224 | 2598 |
| Gemini-2.5-Pro | 31.94 | 22.22 | 23.61 | 77.78 | 51.33 | 71.80 | 55.84 | 65.23 | 79.02 | 95.90 | 44.46 | 82.39 | 32.64 | 16.24 | 2192 | 1963 | 2028 | 2308 | 2551 |
| GPT-O3-Mini-High | 26.39 | 23.61 | 22.22 | 72.22 | 47.69 | 64.28 | 53.07 | 50.76 | 75.79 | 94.67 | 44.19 | 84.51 | 34.03 | 11.11 | 2088 | 1807 | 1894 | 2284 | 2449 |
| Gemini-2.5-Flash | 15.28 | 23.61 | 23.61 | 62.50 | 41.29 | 56.81 | 43.48 | 43.79 | 69.89 | 93.97 | 36.06 | 75.35 | 22.92 | 6.84 | 1945 | 1700 | 1700 | 2091 | 2505 |
| Claude-Sonnet-4.5 | 11.11 | 20.83 | 34.72 | 66.67 | 38.30 | 53.08 | 36.69 | 49.80 | 65.79 | 85.63 | 27.05 | 61.97 | 9.72 | 5.13 | 1848 | 1766 | 1595 | 2057 | 2060 |
| GPT-4.1 | 4.17 | 13.89 | 22.22 | 40.28 | 24.78 | 35.99 | 21.47 | 24.90 | 47.56 | 79.73 | 18.32 | 48.59 | 4.86 | 3.42 | 1482 | 1339 | 1134 | 1724 | 1994 |
| **Open-weight Thinking LLMs** | | | | | | | | | | | | | | | | | | | |
| GPT-OSS-120B-High | 50.00 | 26.39 | 11.11 | 87.50 | 62.78 | 72.88 | 69.46 | 63.36 | 80.92 | 96.82 | 60.14 | 92.25 | 54.86 | 30.77 | 2205 | 1950 | 2122 | 2264 | 2520 |
| GPT-OSS-20B-High | 22.22 | 29.17 | 23.61 | 75.00 | 49.55 | 57.72 | 47.18 | 44.47 | 72.55 | 94.07 | 52.81 | 86.62 | 42.36 | 17.95 | 2020 | 1763 | 1797 | 2167 | 2504 |
| GPT-OSS-120B | 29.17 | 23.61 | 20.83 | 73.61 | 49.23 | 59.90 | 53.89 | 43.75 | 72.87 | 91.14 | 47.78 | 87.32 | 33.33 | 19.66 | 2032 | 1638 | 1894 | 2193 | 2493 |
| GPT-OSS-20B | 19.44 | 23.61 | 25.00 | 68.06 | 42.36 | 53.94 | 48.43 | 37.94 | 66.58 | 88.26 | 42.80 | 81.69 | 29.86 | 12.82 | 1901 | 1501 | 1660 | 2165 | 2383 |
| Qwen3-32B | 9.72 | 15.28 | 29.17 | 54.17 | 32.86 | 42.00 | 31.36 | 31.17 | 56.64 | 81.73 | 27.70 | 67.61 | 6.25 | 6.84 | 1665 | 1342 | 1455 | 1959 | 2022 |
| DeepSeek-R1 | 6.94 | 19.44 | 26.39 | 52.78 | 33.43 | 42.29 | 30.22 | 27.05 | 55.89 | 80.92 | 28.87 | 69.01 | 8.33 | 6.84 | 1617 | 1443 | 1278 | 1906 | 2015 |
| Qwen3-14B | 5.56 | 15.28 | 25.00 | 45.83 | 27.24 | 34.59 | 24.51 | 25.66 | 47.57 | 74.78 | 22.73 | 58.45 | 5.90 | 7.69 | 1402 | 976 | 1241 | 1652 | 1938 |
| QWQ-32B | 5.56 | 13.89 | 26.39 | 45.83 | 26.56 | 33.84 | 19.15 | 26.44 | 49.93 | 71.94 | 23.95 | 57.75 | 18.31 | 11.97 | 1491 | 1281 | 1113 | 1877 | 1956 |
| Qwen3-30B | 5.56 | 20.83 | 18.06 | 44.44 | 27.68 | 36.69 | 24.94 | 27.25 | 47.33 | 72.15 | 23.18 | 63.38 | 14.93 | 7.26 | 1549 | 1201 | 1323 | 1862 | 1995 |
| Qwen3-8B | 1.39 | 12.50 | 26.39 | 40.28 | 24.25 | 31.03 | 25.37 | 23.23 | 40.68 | 69.90 | 19.05 | 53.52 | 15.56 | 7.26 | 1426 | 1206 | 1312 | 1534 | 1789 |
| DeepSeek-R1-Distill-Llama-70B | 1.39 | 8.33 | 23.61 | 33.33 | 20.50 | 32.30 | 22.28 | 23.63 | 39.29 | 76.89 | 16.88 | 43.66 | 15.85 | 7.69 | 1283 | 1042 | 1103 | 1472 | 1665 |
| DeepSeek-R1-Distill-Qwen-32B | 1.39 | 8.33 | 20.83 | 30.56 | 19.14 | 27.03 | 11.43 | 15.59 | 37.55 | 44.75 | 14.86 | 40.14 | 1.39 | 1.71 | 1284 | 964 | 1074 | 1631 | 1549 |
| Qwen3-4B | 1.39 | 8.33 | 16.67 | 26.39 | 16.81 | 24.28 | 18.35 | 16.17 | 29.91 | 54.21 | 13.61 | 31.08 | 12.60 | 5.13 | 1153 | 970 | 897 | 1332 | 1622 |
| DeepSeek-R1-Distill-Qwen-14B | 1.39 | 2.78 | 9.72 | 13.89 | 13.41 | 22.77 | 15.33 | 15.23 | 25.64 | 47.24 | 10.56 | 25.35 | 9.15 | 4.05 | 1089 | 897 | 991 | 1166 | 1457 |
| DeepSeek-R1-Distill-Llama-8B | 0.00 | 0.00 | 2.78 | 2.78 | 3.10 | 11.86 | 10.82 | 10.37 | 9.51 | 33.46 | 2.46 | 10.56 | 1.56 | 0.00 | 724 | 724 | 628 | 705 | 1103 |
| **Open-weight Non-Thinking LLMs** | | | | | | | | | | | | | | | | | | | |
| DeepSeek-V3 | 4.17 | 8.33 | 22.22 | 34.72 | 21.70 | 31.76 | 16.42 | 20.86 | 41.88 | 73.73 | 17.10 | 43.66 | 2.78 | 2.56 | 1283 | 1239 | 1187 | 1598 | 1827 |
| Qwen3-32B-Non-Thinking | 1.39 | 4.17 | 11.11 | 16.67 | 12.92 | 24.64 | 14.14 | 17.35 | 24.10 | 52.37 | 8.78 | 21.13 | 0.69 | 4.27 | 1040 | 957 | 844 | 1227 | 1251 |
| Qwen2.5-Coder-32B-Instruct | 1.39 | 2.78 | 9.72 | 13.89 | 11.25 | 19.90 | 15.31 | 17.26 | 22.73 | 48.38 | 6.15 | 22.54 | 5.90 | 1.71 | 1023 | 983 | 701 | 1247 | 1384 |
| Qwen2.5-Coder-14B-Instruct | 1.39 | 2.78 | 6.94 | 11.11 | 9.66 | 19.56 | 13.48 | 14.56 | 23.55 | 46.20 | 5.53 | 20.42 | 4.51 | 1.71 | 966 | 935 | 849 | 969 | 1360 |
| Mistral-Large-Instruct-2411 | 1.39 | 1.39 | 8.33 | 11.11 | 9.99 | 18.70 | 14.62 | 15.13 | 22.66 | 43.03 | 5.90 | 21.13 | 4.65 | 1.71 | 1023 | 939 | 875 | 1122 | 1376 |
| Mistral-Small-3.1-24B-2503 | 1.39 | 0.00 | 9.72 | 11.11 | 7.75 | 19.08 | 10.67 | 10.92 | 20.04 | 42.86 | 4.75 | 17.61 | 3.41 | 1.71 | 909 | 805 | 822 | 879 | 1334 |
| Llama-4-Scout | 1.39 | 1.39 | 5.56 | 8.33 | 9.88 | 19.60 | 10.67 | 11.97 | 24.11 | 47.09 | 6.32 | 21.83 | 4.34 | 1.71 | 1008 | 825 | 892 | 1107 | 1316 |
| Qwen2.5-72B | 1.39 | 2.78 | 5.56 | 9.72 | 9.90 | 19.24 | 10.84 | 12.99 | 22.96 | 45.43 | 5.55 | 14.08 | 0.00 | 1.71 | 1000 | 875 | 862 | 1022 | 1508 |
| Llama-3.3-70B-Instruct | 0.00 | 1.39 | 8.33 | 9.72 | 10.00 | 21.37 | 15.04 | 16.67 | 26.37 | 50.32 | 5.65 | 20.42 | 4.79 | 1.71 | 1056 | 899 | 1069 | 1020 | 1458 |
| Qwen3-30B-Non-Thinking | 1.39 | 0.00 | 6.94 | 8.33 | 10.48 | 17.28 | 10.60 | 12.11 | 22.83 | 43.92 | 6.99 | 23.94 | 5.49 | 1.71 | 989 | 962 | 791 | 1052 | 1425 |
| Qwen3-4B-Non-Thinking | 0.00 | 1.39 | 5.56 | 6.94 | 6.65 | 15.30 | 6.65 | 8.76 | 10.91 | 44.36 | 4.47 | 10.56 | 0.00 | 2.56 | 894 | 818 | 753 | 932 | 1303 |
| Qwen3-8B-Non-Thinking | 0.00 | 1.39 | 2.78 | 4.17 | 7.53 | 16.82 | 9.78 | 9.01 | 14.16 | 39.65 | 4.04 | 10.56 | 0.00 | 1.71 | 843 | 745 | 701 | 842 | 1357 |
| CODESTRAL-22B-V0.1 | 0.00 | 1.39 | 2.78 | 4.17 | 6.84 | 15.94 | 9.81 | 7.46 | 6.86 | 42.39 | 4.34 | 8.45 | 1.39 | 2.56 | 912 | 948 | 784 | 895 | 1275 |
| Llama-3.1-8B-Instruct | 0.00 | 1.39 | 1.39 | 2.78 | 4.19 | 13.49 | 7.90 | 6.15 | 7.60 | 30.13 | 2.45 | 5.63 | 0.00 | 0.85 | 761 | 714 | 644 | 808 | 1073 |

Table A9: Main results of all models evaluated on 72 OI-Bench contests, with expanded Human Percentile, Pass Rate, and CF rating breakdowns.

| Model | IM | MA | AH | PS | SO | GR | GTR | BS | NT | GT | DS | CB | DP | TR | ST |
|---|---|---|---|---|---|---|---|---|---|---|---|---|---|---|---|
| **Proprietary LLMs** | | | | | | | | | | | | | | | |
| GPT-5 | 71.79 | 71.43 | 43.48 | 73.33 | 75.56 | 60.00 | 71.43 | 54.84 | 64.71 | 66.67 | 66.27 | 64.71 | 46.88 | 37.50 | 56.41 |
| Gemini-2.5-Pro | 66.67 | 71.43 | 30.43 | 53.33 | 57.78 | 37.14 | 42.86 | 38.71 | 35.29 | 44.44 | 38.55 | 58.82 | 23.44 | 20.83 | 30.77 |
| GPT-O3-Mini-High | 64.10 | 71.43 | 34.78 | 46.67 | 60.00 | 37.14 | 46.43 | 41.94 | 41.18 | 38.89 | 38.55 | 47.06 | 34.38 | 20.83 | 28.21 |
| Gemini-2.5-Flash | 64.10 | 71.43 | 30.43 | 46.67 | 48.89 | 28.57 | 25.00 | 32.26 | 29.41 | 29.63 | 30.12 | 47.06 | 20.31 | 12.50 | 15.38 |
| GPT-4.1 | 53.85 | 50.00 | 26.09 | 40.00 | 13.33 | 14.29 | 7.14 | 12.90 | 17.65 | 12.96 | 12.05 | 29.41 | 6.25 | 4.17 | 5.13 |
| **Open-source Thinking LLMs** | | | | | | | | | | | | | | | |
| GPT-OSS-120B-High | 71.79 | 71.43 | 39.13 | 73.33 | 82.22 | 57.14 | 75.00 | 51.61 | 58.82 | 55.56 | 62.65 | 58.82 | 46.88 | 41.67 | 51.28 |
| GPT-OSS-120B-Medium | 64.10 | 64.29 | 34.78 | 53.33 | 60.00 | 40.00 | 53.57 | 38.71 | 41.18 | 44.44 | 44.58 | 58.82 | 35.94 | 25.00 | 35.90 |
| GPT-OSS-120B-Low | 61.54 | 71.43 | 30.43 | 46.67 | 37.78 | 31.43 | 35.71 | 29.03 | 23.53 | 27.78 | 27.71 | 47.06 | 17.19 | 16.67 | 15.38 |
| GPT-OSS-20B-High | 69.44 | 76.92 | 50.00 | 64.29 | 73.81 | 53.57 | 51.85 | 48.15 | 46.67 | 44.23 | 50.70 | 53.33 | 50.00 | 40.00 | 48.48 |
| GPT-OSS-20B-Medium | 63.16 | 71.43 | 40.91 | 57.14 | 51.11 | 36.36 | 35.71 | 36.67 | 47.06 | 30.19 | 36.59 | 66.67 | 29.69 | 22.73 | 26.32 |
| GPT-OSS-20B-Low | 56.41 | 64.29 | 30.43 | 40.00 | 33.33 | 17.14 | 25.00 | 23.33 | 23.53 | 27.78 | 24.69 | 35.29 | 17.46 | 12.50 | 13.16 |
| Seed-OSS | 61.54 | 64.29 | 36.36 | 53.33 | 48.89 | 31.43 | 32.14 | 38.71 | 35.29 | 27.78 | 34.94 | 52.94 | 26.56 | 12.50 | 28.21 |
| Qwen3-32B | 58.97 | 61.54 | 30.43 | 35.71 | 28.89 | 21.88 | 21.43 | 16.67 | 29.41 | 22.64 | 22.22 | 29.41 | 14.29 | 4.35 | 8.11 |
| DeepSeek-R1 | 61.54 | 64.29 | 30.43 | 33.33 | 28.89 | 17.14 | 17.86 | 22.58 | 29.41 | 22.22 | 20.48 | 29.41 | 15.62 | 4.17 | 7.69 |
| Qwen3-14B | 51.28 | 61.54 | 26.09 | 35.71 | 24.44 | 15.62 | 14.29 | 13.33 | 29.41 | 18.87 | 19.75 | 35.29 | 12.70 | 4.35 | 5.41 |
| QWQ-32B | 53.85 | 61.54 | 26.09 | 28.57 | 26.67 | 15.62 | 10.71 | 20.00 | 23.53 | 15.09 | 13.58 | 29.41 | 14.29 | 4.35 | 5.41 |
| Qwen3-30B | 43.59 | 61.54 | 26.09 | 28.57 | 31.11 | 18.75 | 28.57 | 13.33 | 29.41 | 24.53 | 23.46 | 41.18 | 15.87 | 4.35 | 5.41 |
| Qwen3-8B | 33.33 | 57.14 | 17.39 | 26.67 | 8.89 | 5.71 | 0.00 | 9.68 | 29.41 | 9.26 | 13.25 | 35.29 | 10.94 | 4.17 | 2.56 |
| DeepSeek-R1-Distill-Llama-70B | 41.03 | 50.00 | 17.39 | 20.00 | 20.00 | 17.14 | 10.71 | 16.13 | 17.65 | 14.81 | 13.25 | 11.76 | 9.38 | 4.17 | 5.13 |
| DeepSeek-R1-Distill-Qwen-32B | 38.46 | 46.15 | 21.74 | 14.29 | 15.56 | 12.50 | 7.14 | 10.00 | 11.76 | 5.66 | 8.64 | 11.76 | 3.17 | 0.00 | 2.70 |
| Qwen3-4B | 46.15 | 50.00 | 17.39 | 13.33 | 15.56 | 8.57 | 10.71 | 9.68 | 11.76 | 11.11 | 9.64 | 17.65 | 4.69 | 4.17 | 2.56 |
| DeepSeek-R1-Distill-Qwen-14B | 33.33 | 46.15 | 8.70 | 0.00 | 13.33 | 6.25 | 7.14 | 10.00 | 5.88 | 9.43 | 6.17 | 5.88 | 1.59 | 4.35 | 0.00 |
| DeepSeek-R1-Distill-Llama-8B | 12.82 | 23.08 | 0.00 | 0.00 | 0.00 | 0.00 | 0.00 | 0.00 | 0.00 | 0.00 | 0.00 | 0.00 | 0.00 | 0.00 | 0.00 |
| **Open-source Non-Thinking LLMs** | | | | | | | | | | | | | | | |
| DeepSeek-V3 | 51.28 | 46.15 | 21.74 | 28.57 | 20.00 | 12.50 | 14.29 | 13.33 | 17.65 | 15.09 | 14.81 | 11.76 | 7.94 | 8.70 | 8.11 |
| Qwen3-32B-Non-Thinking | 25.64 | 42.86 | 13.04 | 0.00 | 6.67 | 5.71 | 3.57 | 9.68 | 11.76 | 7.41 | 2.41 | 11.76 | 4.69 | 0.00 | 2.56 |
| Qwen2.5-Coder-32B-Instruct | 25.64 | 46.15 | 8.70 | 0.00 | 6.67 | 6.25 | 3.57 | 6.67 | 11.76 | 3.77 | 4.94 | 5.88 | 3.17 | 0.00 | 0.00 |
| Qwen2.5-Coder-14B-Instruct | 20.51 | 46.15 | 9.09 | 0.00 | 6.82 | 3.12 | 3.57 | 3.33 | 5.88 | 5.77 | 1.23 | 11.76 | 1.61 | 0.00 | 0.00 |
| Mistral-Large-Instruct-2411 | 28.21 | 42.86 | 13.04 | 0.00 | 4.44 | 0.00 | 3.57 | 9.68 | 5.88 | 3.70 | 1.20 | 11.76 | 3.12 | 0.00 | 0.00 |
| Mistral-Small-3.1-24B-2503 | 23.08 | 46.15 | 8.70 | 0.00 | 4.44 | 0.00 | 3.57 | 3.33 | 5.88 | 3.77 | 2.47 | 5.88 | 1.59 | 0.00 | 0.00 |
| Qwen2.5-72B | 23.08 | 38.46 | 9.09 | 0.00 | 6.82 | 3.12 | 3.57 | 6.67 | 5.88 | 1.92 | 2.47 | 0.00 | 1.61 | 0.00 | 0.00 |
| Llama-3.3-70B-Instruct | 23.08 | 38.46 | 8.70 | 0.00 | 6.67 | 3.12 | 3.57 | 3.33 | 5.88 | 5.66 | 1.23 | 5.88 | 1.59 | 0.00 | 0.00 |
| Qwen3-30B-Non-Thinking | 23.68 | 30.77 | 8.70 | 6.67 | 8.89 | 2.86 | 7.14 | 6.45 | 5.88 | 9.43 | 2.44 | 17.65 | 0.00 | 0.00 | 0.00 |
| Qwen3-4B-Non-Thinking | 28.21 | 42.86 | 8.70 | 0.00 | 4.44 | 0.00 | 3.57 | 6.45 | 5.88 | 5.56 | 1.20 | 5.88 | 1.56 | 0.00 | 0.00 |
| Qwen3-8B-Non-Thinking | 20.51 | 30.77 | 4.35 | 0.00 | 8.89 | 2.86 | 0.00 | 3.23 | 5.88 | 0.00 | 1.20 | 0.00 | 0.00 | 0.00 | 0.00 |
| Codestral-22B-V0.1 | 20.51 | 38.46 | 4.35 | 0.00 | 2.22 | 0.00 | 3.57 | 0.00 | 0.00 | 7.55 | 0.00 | 0.00 | 1.59 | 0.00 | 0.00 |
| Llama-3.1-8B-Instruct | 15.38 | 38.46 | 4.35 | 0.00 | 2.22 | 0.00 | 3.57 | 3.33 | 5.88 | 1.89 | 1.23 | 0.00 | 0.00 | 0.00 | 0.00 |
| Qwen3-14B-Non-Thinking | 18.75 | 18.18 | 9.52 | 0.00 | 13.95 | 5.88 | 7.69 | 6.45 | 6.25 | 5.88 | 3.75 | 0.00 | 1.61 | 0.00 | 0.00 |
| DeepSeek-Coder-V2-Lite-Instruct | 12.82 | 30.77 | 0.00 | 0.00 | 0.00 | 0.00 | 3.57 | 3.33 | 0.00 | 3.77 | 0.00 | 0.00 | 0.00 | 0.00 | 0.00 |
| Qwen2.5-Coder-7B-Instruct | 13.16 | 35.71 | 8.70 | 0.00 | 2.22 | 0.00 | 3.57 | 3.45 | 5.88 | 2.04 | 1.23 | 0.00 | 1.59 | 0.00 | 0.00 |

Table A10: Pass rate of all tags for each model, from easiest to hardest based on difficulty labels. Abbreviations: IM (implementation), MA (mathematics), AH (ad-hoc), PS (prefix sum), SO (sorting), GR (greedy), GTR (graph traversal), BS (binary search), NT (number theory), GT (graph theory), DS (data structures), CB (combinatorics), DP (dynamic programming), TR (tree), ST (segment tree).

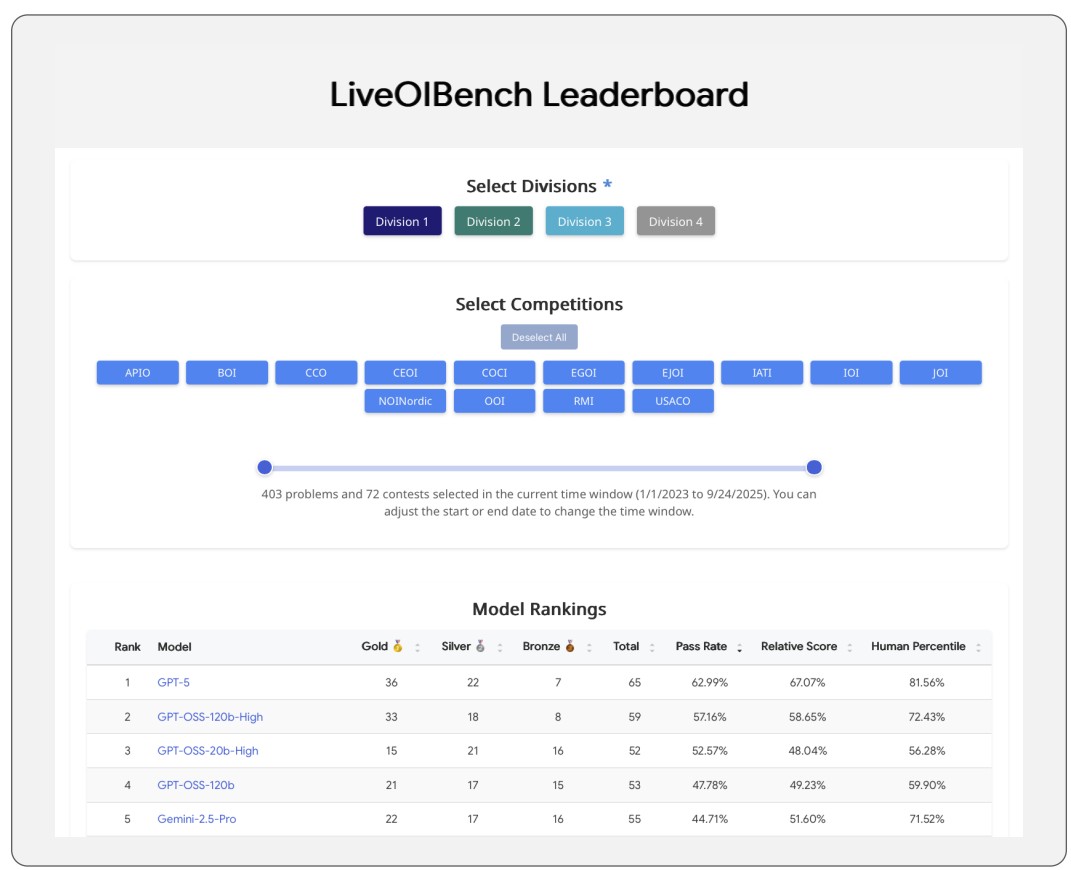

Figure A1: LiveOIBench website that displays leaderboard across models

## F  ADDITIONAL ANALYSIS

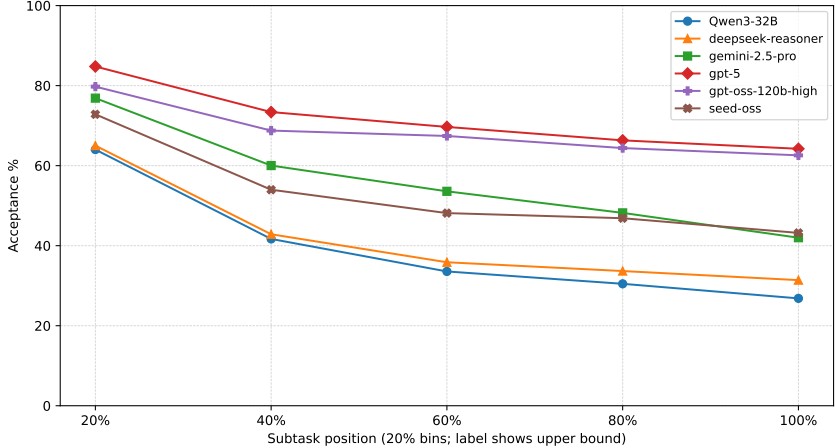

Figure A2: Mainstream model performance over sub-task positions. As expected, later sub-tasks poses greater challenges for LLMs to tackle.

### F.1  MODEL PERFORMANCE ACROSS YEARS

Figure A3 shows quarterly pass rates of four mainstream LLMs from Q4'22 to Q2'25. The performance trends are broadly similar across models: all experience an early decline in 2023, recover through 2024, peak around late 2024 to early 2025, and then drop again in Q2'25. Importantly, there is no sharp bump or drop around the knowledge cutoff, suggesting that these models are not facing

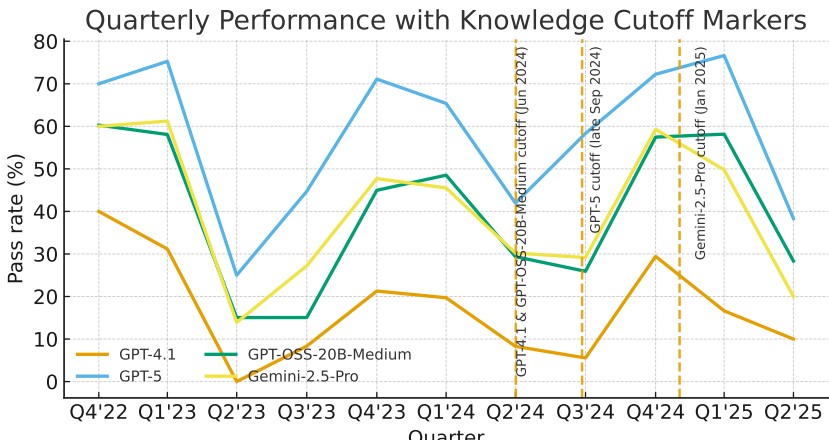

Figure A3: Mainstream model performance over quarters. This plot shows consistent performance trend among select models, as well as confirms no data contamination in mainstream LLMs.

significant data contamination issues. Quantitatively, GPT-5 consistently leads: in its stronger quarters (Q1'23, Q4'23, Q1'25), it consistently outperforms Gemini-2.5-Pro and GPT-OSS-20B-Medium by about $15 \textasciitilde 25$ percentage points, which is in line with A9.

## F.2 INFERENCE-TIME SCALING

Inference-time scaling has been shown effective for improving model performance in math (Snell et al., 2024; Brown et al., 2024) and coding (Li et al., 2025a; Ehrlich et al., 2025) domains. We investigate two dimensions: *parallel scaling* involves sampling multiple diverse solution candidates (Chen et al., 2021b; Jain et al., 2024), while *sequential scaling* generates long chains-of-thought with complex reasoning strategies such as self-reflection and backtracking (DeepSeek-AI et al., 2025).

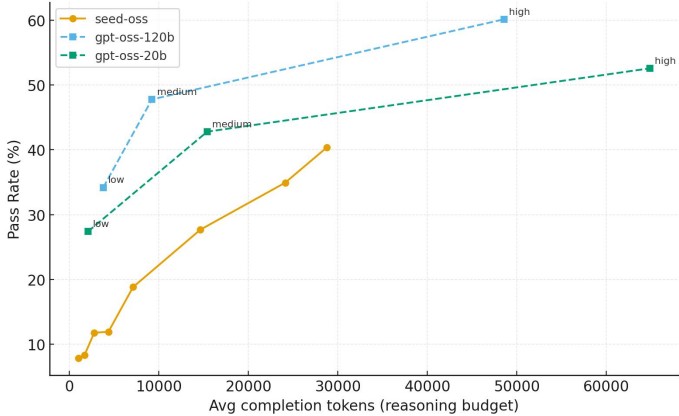

Figure A4: **Sequential Scaling** plots the pass rate against the reasoning budget (measured in average completion tokens), showing that performance improves with more extensive reasoning, though models exhibit different token efficiencies.

**Parallel Scaling: GPT-5 demonstrates superior coding capacity boundary.** Figure 2 reveals significant differences in coding capacity boundaries (Yue et al., 2025) across models as measured by pass@k. GPT-5 could pass around $64\%$ of the problems give 8 attempts per problem. The steepest improvements occur between Pass@1 and Pass@4, indicating that the marginal benefit of additional attempts diminishes rapidly as models approach their capacity limits (Kulal et al., 2019). The persistent performance gaps between proprietary and open-source models across all sampling levels suggest fundamental differences in maximum coding capability rather than artifacts of insufficient attempts (Li et al., 2022b; Hendrycks et al., 2021a).

**Sequential Scaling: Reasoning models benefit from additional reasoning token budget.** Figure A4 shows pass rates improving as token budget increases across all three models. GPT-OSS-120B achieves the highest performance with the fewest tokens generated. A key insight emerges: smaller models can approach larger model performance with sufficient reasoning budget, suggesting a practical trade-off for resource-constrained practitioners who may prefer specialized smaller models over large ones.

Both scaling approaches provide complementary benefits but face efficiency limitations. Sequential scaling shows promise for complex algorithmic problems but requires substantial computational resources, while parallel scaling reveals each model's performance ceiling as improvements plateau with additional samples (Chen et al., 2021b; Austin et al., 2021). Future work could focus on developing hybrid approaches that combine both scaling paradigms while reducing computational overhead.

| Model | Relative Score (%) | Pass Rate (%) | Gold (%) | Silver (%) | Bronze (%) | Total Medals (%) |
|---|---|---|---|---|---|---|
| GPT-OSS-120B_Python | 44.35 | 46.97 | 13.64 | 13.64 | 27.27 | 54.55 |
| GPT-OSS-120B_Java | 50.78 | 56.82 | 9.09 | 36.36 | 27.27 | 72.73 |
| GPT-OSS-120B_C++ | 53.26 | 59.09 | 27.27 | 31.82 | 13.64 | 72.73 |

Table A11: Performance comparison of GPT-OSS-120B across Python, Java, and C++ on 22 USACO competitive programming contests. Metrics shown include relative score, pass rate, and medal distributions (gold, silver, bronze, and total medal percentage). Results indicate that C++ achieves the highest overall performance, followed by Java, with Python exhibiting significantly lower performance, likely due to differences in execution speed, memory management, and efficiency of standard libraries.

### F.3 PERFORMANCE VARIATIONS ACROSS PROGRAMMING LANGUAGES

To provide deeper insights into language-specific model capabilities, we present a comparative analysis of GPT-OSS-120B's performance across Python, Java, and C++, evaluated on a subset of 132 problems from the USACO contest series. As shown in the Table A11, the model performs best when generating solutions in C++, highlighting the inherent advantage of C++ in competitive programming due to its superior execution speed, precise memory control, and optimized functions. Java delivers intermediate performance; although robust and widely supported, it incurs overhead from JVM execution and automated memory management, placing it slightly behind C++. Meanwhile, although Python is the most resourceful and extensively studied language for evaluating LLM coding capability, it shows a considerable performance gap due to its interpreted nature and dynamic typing, sacrificing critical speed and efficiency. Thus, contest results clearly show C++ performing best, followed by Java, with Python significantly behind.

To better understand the reasons behind these performance differences, we further analyzed GPT-OSS-120B's reasoning behaviors across these programming languages. As illustrated in the Figure A5, the model consistently allocates reasoning tokens similarly across Python, Java, and C++, indicating that its reasoning strategy remains stable irrespective of language choice. This stability in reasoning behavior suggests that the performance advantage observed for C++ primarily stems from computational efficiency rather than differences in the underlying reasoning approach.

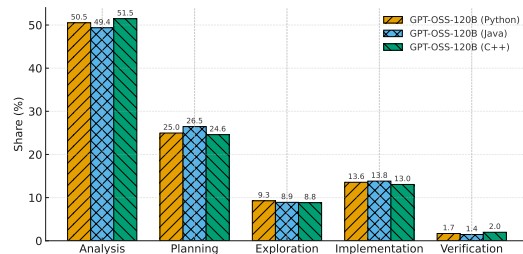

Figure A5: Histogram showing similarity scores between accepted model solutions (GPT-5 and Grok-4-Fast-Reasoning) and their corresponding official solutions. All similarity scores remain below 50%, indicating that accepted model solutions are substantially distinct from official solutions.

### F.4 IMPACT OF TEMPLATE SIMILARITY ON MODEL PERFORMANCE

To investigate if models exhibit similar performance on problems sharing similar solution templates, we compute the official solution similarity for each problem pair and analyze its relationship with the models' performance differences on these problems. Specifically, we assessed GPT-5 and

Grok-4-fast-reasoning by plotting the absolute difference in their relative scores against each problem pair's solution similarity (Figures A6). If template-based contamination significantly influenced model performance, we would expect to observe consistently small performance differences between problem pairs with similar official solutions. However, our analysis reveals only a very weak and negligible correlation (GPT-5: Pearson's r = -0.084; Grok-4: Pearson's r = -0.072). This indicates that even problem pairs with similar official solution structures often result in substantially different model performances. Additionally, we found that most problem pairs exhibit low similarity, emphasizing the wide variety of problem-solving approaches necessary for success on our benchmark.

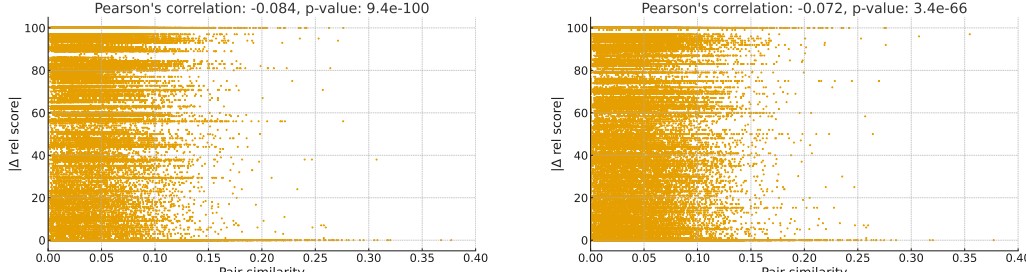

Figure A6: Scatter plots of solution similarity vs. model performance differences for GPT-5 (left) and Grok-4 (right). Weak correlations (GPT-5: $r = -0.08$; Grok-4: $r = -0.07$) indicate minimal template-based contamination.

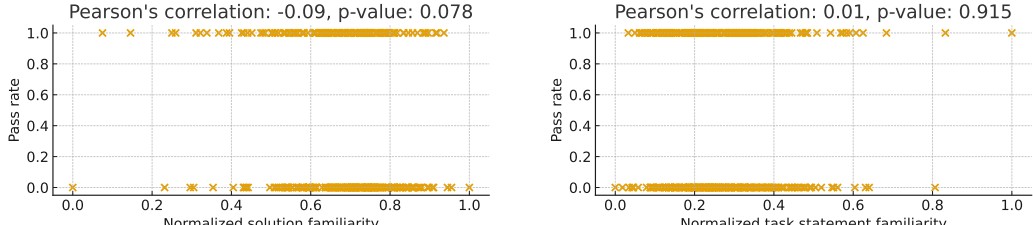

Figure A7: No significant positive correlation is observed between GPT-OSS-120B's familiarity with task statements and solutions (normalized via min-max scaling) and its performance, indicating that higher familiarity does not necessarily translate to better outcomes.

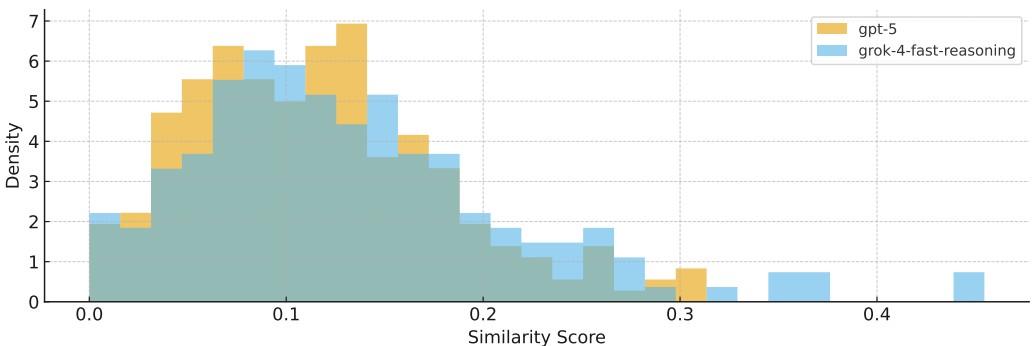

Figure A8: Histogram showing similarity scores between accepted model solutions (GPT-5 and Grok-4-Fast-Reasoning) and their corresponding official solutions. All similarity scores remain below 50%, indicating that accepted model solutions are substantially distinct from official solutions. Density represents the probability per unit similarity, so that the total area of the histogram equals 1.

## F.5 MORE ANALYSIS ON HOW DIFFERENT ATTRIBUTES CAN IMPACT THE MODEL PERFORMANCE

In Table A10 and Table A12, we present a detailed analysis of model performance across various algorithmic categories, examining three primary factors: model size, reasoning-token budget, and training strategies (RL and distillation).

| Model (%) | IM | MA | AH | PS | SO | GR | GTR | BS | NT | GT | DS | CB | DP | TR | ST | Avg |
|---|---|---|---|---|---|---|---|---|---|---|---|---|---|---|---|---|
| GPT-20B-Low → GPT-120B-Low | +9.10 | +11.10 | +0.00 | +16.70 | +13.40 | +83.30 | +42.80 | +24.40 | +0.00 | +0.00 | +12.20 | +33.40 | -1.50 | +33.40 | +16.90 | **18.20** |
| GPT-20B-Low → GPT-20B-High | +23.06 | +19.64 | +64.28 | +60.73 | +121.44 | +212.64 | +107.40 | +106.46 | +98.36 | +59.21 | +105.40 | +51.09 | +186.43 | +220.00 | +268.33 | **113.47** |
| Qwen3-14B → GPT-20B-Medium | +23.17 | +16.07 | +56.77 | +59.94 | +109.08 | +132.79 | +149.86 | +174.99 | +60.00 | +60.00 | +85.19 | +88.91 | +133.46 | +322.07 | +286.33 | **118.06** |

Table A12: Percentage improvements across algorithmic task categories from scaling model parameters and reasoning-token budgets.

| Model | Relative Score (%) | Pass Rate (%) | Human Percentile |
|---|---|---|---|
| GPT-OSS-120B-P1 | 47.76 | 45.85 | 58.50 |
| GPT-OSS-120B-P2 | 48.07 | 47.10 | 59.76 |
| GPT-OSS-120B-P3 | 46.75 | 45.69 | 57.73 |
| GPT-OSS-120B-P4 | 48.04 | 45.68 | 59.49 |
| GPT-OSS-120B-Medium | 48.88 | 47.33 | 58.10 |
| **Mean ± Std** | **47.90 ± 0.69** | **46.33 ± 0.73** | **58.72 ± 0.79** |

Table A13: Performance of GPT-OSS-120B-Medium under different prompt variants.

**Model Size**: Scaling GPT-OSS from 20B to 120B parameters under low-reasoning effort yields an average improvement of +18%. This gain is concentrated mainly in Greedy (+83.30%), Graph Traversal (+42.80%), Combinatorics (+33.40%), and Tree (+33.40%). These patterns indicate that additional parameters enhance the model's structural representations and pattern-recognition capacity, enabling better handling of connectivity and global relational structure.

**Reasoning Tokens**: Increasing the reasoning-token budget from low to high, while maintaining the same GPT-OSS-20B model weights, produces a substantially larger average improvement of +113%. Gains are most pronounced in algorithmic tasks such as string manipulation (+268.33%), tree traversal (+220.00%), greedy algorithms (+212.64%), dynamic programming (+186.43%), and sorting (+121.44%), all of which demand extended multi-step, sequential reasoning. Notably, greedy and tree traversal appear in the top-5 for both factors: larger models recognize their structural patterns more reliably, while deeper reasoning is essential to execute the multi-step, stateful chains these problems demand.

**RL vs. Distillation**: Since each model family differs in architecture, pretraining data, and training pipelines, it's difficult to cleanly isolate the effects of RL versus distillation on downstream performance. However, preliminary comparisons between Qwen3-14B (a distilled model) and GPT-OSS-20B (trained via SFT and RL) suggest RL-trained models, with SFT warmup and enough RL training, can substantially outperform distilled models, with an average improvement of +118%. Isolating the contribution of each training stage is an important future direction for us to understand how these methods shape algorithmic skills.

Overall, our analyses show that reasoning token allocation, far more than model size, drives improvements on the most algorithmically complex tasks. Also, directly trained RL models perform better than distilled models.

## F.6 PROMPT SENSITIVITY

We generate four alternative prompt variants using GPT-5 and run GPT-OSS-120B-Medium with four additional prompt variants to explore how different instructions might influence the model's performance. While these prompts do not represent all possible instruction styles, we observe minimal performance differences across them, suggesting that prompt variants have little influence on model performance. The default prompt and prompt variants can be found at C and F.7.

In Table A13, across all five prompt variants, GPT-OSS-120B achieves $47.9 \pm 0.7\%$ relative score and $46.3 \pm 0.7\%$ pass rate, with human percentile tightly clustered around $58.7 \pm 0.8$, indicating low sensitivity to prompt choices. A more systematic and large-scale study of prompt sensitivity is an interesting direction for future work.

### F.7 ANALYSIS OF TEMPLATE-BASED CONTAMINATION AND MODEL PERFORMANCE

We selected CodeContest (Li et al., 2022a; Wang et al., 2025b) as our seed dataset to represent template solutions commonly found in training data, as it is among the most prominent competitive coding datasets with over 10,000 problems. For each problem, we sampled three C++ solutions, filtering out solutions shorter than 20 lines or longer than 1,000 lines. Using Dolos, we calculated the similarity between official solutions and these template solutions, recording the highest similarity score for each official solution. A higher similarity score indicates a greater likelihood of template-based contamination.

To examine whether template similarity influences model performance, we conducted correlation analyses using GPT-5 and Grok-4-Fast-Reasoning. In Figure A9, our analysis revealed weak correlations between template similarity and relative performance scores: GPT-5 exhibited a Pearson correlation coefficient of 0.136, and Grok-4 demonstrated a Pearson correlation coefficient of 0.193. These correlations, though slightly stronger than those observed in our previous analysis, remain limited.

This finding indicates that template similarity contributes minimally to model performance. Rather, the dominant factor influencing performance is the model's intrinsic problem-solving and reasoning capabilities. Given the nature of competitive coding problems, models rely on learned algorithmic strategies and analytical techniques to solve problems effectively. Therefore, we contend that the minimal template-based contamination observed does not undermine the validity of our evaluations. Models must effectively utilize learned skills from their training, applying them innovatively to solve new and complex challenges.

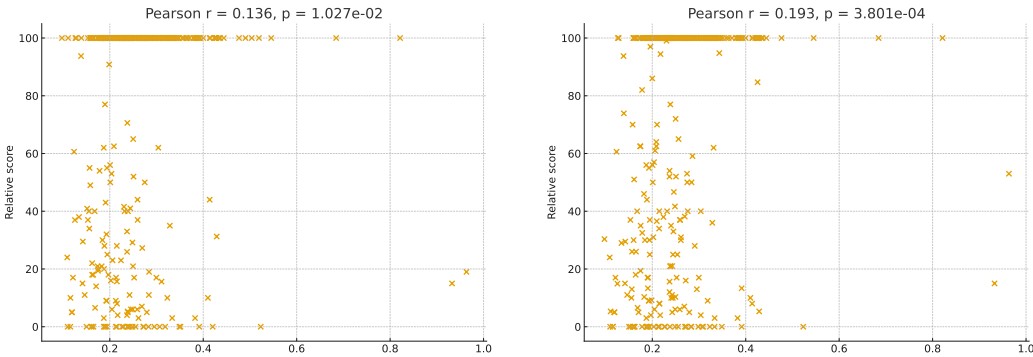

Figure A9: Scatter plots illustrating the relationship between official-solution similarity and relative scores for GPT-5 (left) and Grok-4-Fast-Reasoning (right). Both models exhibit weak correlations (Pearson r = 0.136, p = 0.010 for GPT-5; Pearson r = 0.193, p < 0.001 for Grok-4), suggesting minimal impact of template similarity on overall model performance.

---

**Prompt Variants**

**Prompt 1.** You are a helpful and knowledgeable AI assistant. Answer the user's questions clearly and concisely, using correct reasoning and accurate information.

**Prompt 2.** You are an expert IOI competitive programming assistant. For each problem, output only a single complete C++17 solution that compiles and solves the task. Do not include explanations, comments, or any text outside one `cpp` code block. Read input from standard input and write output to standard output.

**Prompt 3.** You are an expert IOI problem solver. First, think through the solution step by step and explain your reasoning in a few clear paragraphs or bullet points. After that, output a single complete C++17 solution inside one `cpp` code block that follows your reasoning, reads from standard input, and writes to standard output.

**Prompt 4.** You are a meticulous and safe IOI problem-solving assistant. Always prioritize correctness, handling of corner cases, and numerical stability over constant-factor performance. Carefully verify your reasoning before producing an answer. When writing code, output a single complete C++17 solution in a `cpp` code block, making sure it compiles, handles invalid or extreme inputs robustly when appropriate, and does not perform unsafe or undefined operations.

### F.8    REASONING BEHAVIORS ANALYSIS

As described in Section 5.2, we partition each reasoning trace into segments of approximately 5k tokens, estimated by dividing the total token length by four. We categorize models' reasoning traces into eight behaviors, which we group into five broader categories: **Analysis** (Algorithm/Proof Analysis, Complexity Analysis), **Planning** (Problem Restatement, Subgoal Setting), **Exploration** (Backtracking, Dead-end Recognition), **Implementation** (Pseudo Implementation), and **Validation** (Test Case Verification).

The following prompts were used to elicit and analyze the reasoning behaviors of each segment:

- PR_PROMPT → Problem Restatement (Planning). See Prompt 1.
- CMP_PROMPT → Complexity Analysis (Analysis). See Prompt 2.
- VT_PROMPT → Test Case Verification (Validation). See Prompt 3.
- SUB_PROMPT → Subgoal Setting (Planning). See Prompt 4.
- DED_PROMPT → Dead-end Recognition (Exploration). See Prompt 5.
- BKT_PROMPT → Backtracking (Exploration). See Prompt 6.
- AP_PROMPT → Algorithm/Proof Analysis (Analysis). See Prompt 7.
- PSD_PROMPT → Pseudo Implementation (Implementation). See Prompt 8.

```
PR_PROMPT = """
You are an auditor. Count occurrences of the behavior PR (Problem
    Restatement) in a competitive-programming reasoning trace.

DEFINITION (apply strictly)
PR = Expressing the task in the solver's own words to clarify WHAT
    must be computed/decided/constructed (not HOW).
Include: restating the goal/output/validity conditions; clarifying
    what constitutes a correct answer.

COUNT
- Count 1 per PR-labeled step.

OUTPUT (strict JSON ONLY -- no extra text):
{
  "PR": <integer count>,
  "events": [
    {"snippet": "<short quote>", "reason": "<why it matches PR>"}
  ]
}

<TRACE>
{TRACE}
</TRACE>

Analyze the trace and count the occurrences of PR.
"""
```

Prompt 1: PR_PROMPT (Problem Restatement)

```
CMP_PROMPT = """
You are an auditor. Count occurrences of the behavior CMP (Complexity
    Analysis) in a competitive-programming reasoning trace.
DEFINITION
CMP = Analyzing asymptotic time/space complexity and feasibility
    versus constraints.

COUNT
- Count 1 per CMP-labeled step.

OUTPUT (strict JSON ONLY):
{
  "CMP": <integer count>,
  "events": [
    {"snippet": "<short quote>", "reason": "<why it matches CMP>"}
  ]
}

<TRACE>
{TRACE}
</TRACE>

Analyze the trace and count the occurrences of CMP.
"""
```

Prompt 2: CMP_PROMPT (Complexity Analysis)

```
VT_PROMPT = """
You are an auditor. Count occurrences of the behavior V-T (Test Cases
    Verification) in a competitive-programming reasoning trace.

DEFINITION
V-T = Checking the method on specific inputs and comparing with
    expected/reference outcomes.
Include: "On sample 2, expected=5, we get 5"; "Fails on [3,3,2] with
    output 7".

COUNT
- Count 1 per V-T-labeled step (multiple tests in one step = 1).

OUTPUT (strict JSON ONLY):
{
  "V-T": <integer count>,
  "events": [
    {"snippet": "<short quote>", "reason": "<why it matches V-T>"}
  ]
}

<TRACE>
{TRACE}
</TRACE>

Analyze the trace and count the occurrences of V-T.
"""
```

Prompt 3: VT_PROMPT (Test Case Verification)

```
SUB_PROMPT = """
You are an auditor. Count occurrences of the behavior SUB (Subgoal
    Setting) in a competitive-programming reasoning trace.

DEFINITION
SUB = Breaking the solution into intermediate objectives or a
    checklist before implementation.
Include: ordered lists like "parse  preprocess  compute  output";
    milestones like "build graph; find components; count sizes".

COUNT
- Count 1 per SUB-labeled step.

OUTPUT (strict JSON ONLY):
{
  "SUB": <integer count>,
  "events": [
    {"snippet": "<short quote>", "reason": "<why it matches SUB>"}
  ]
}

<TRACE>
{TRACE}
</TRACE>

Analyze the trace and count the occurrences of SUB.
"""
```

Prompt 4: SUB_PROMPT (Subgoal Setting)

```
DED_PROMPT = """
You are an auditor. Count occurrences of the behavior DED (Dead-end
    recognition) in a competitive-programming reasoning trace.
DEFINITION
DED = Explicitly concluding the current approach is incorrect/
    insufficient or cannot meet constraints.
Include: naming a failure mode ("greedy not optimal", "breaks for
    duplicates", "TLE for n=2e5").

COUNT
- Count 1 per DED-labeled step.
OUTPUT (strict JSON ONLY):
{
  "DED": <integer count>,
  "events": [
    {"snippet": "<short quote>", "reason": "<why it matches DED>"}
  ]
}
<TRACE>
{TRACE}
</TRACE>

Analyze the trace and count the occurrences of DED.
"""
```

Prompt 5: DED_PROMPT (Dead-end Recognition)

```
BKT_PROMPT = """
You are an auditor. Count occurrences of the behavior BKT (
    Backtracking) in a competitive-programming reasoning trace.
DEFINITION
BKT = Revising or replacing the plan after recognizing a failure/
    limitation.
Include: "scrap/switch/replace", "instead we will", "new plan: ".
COUNT
- Count 1 per BKT-labeled step.

OUTPUT (strict JSON ONLY):
{
  "BKT": <integer count>,
  "events": [
    {"snippet": "<short quote>", "reason": "<why it matches BKT>"}
  ]
}
<TRACE>
{TRACE}
</TRACE>

Analyze the trace and count the occurrences of BKT.
{TRACE}
"""
```

Prompt 6: BKT_PROMPT (Backtracking)

```
AP_PROMPT = """
You are an auditor. Count occurrences of the behavior AP (Algorithm /
    Proof analysis) in a competitive-programming reasoning trace.

DEFINITION
AP = Justifying WHY the chosen algorithm/structure is correct/
    appropriate (proof sketches, invariants used as correctness
    arguments, reductions implying correctness).
Include: exchange/optimality arguments, loop-invariant proofs,
    reductions with correctness justification, structural reasoning
    that ensures the property.
COUNT
- Count 1 per AP-labeled step.

OUTPUT (strict JSON ONLY):
{
  "AP": <integer count>,
  "events": [
    {"snippet": "<short quote>", "reason": "<why it matches AP>"}
  ]
}
<TRACE>
{TRACE}
</TRACE>

Analyze the trace and count the occurrences of AP.
{TRACE}
"""
```

Prompt 7: AP_PROMPT (Algorithm/Proof Analysis)

```
PSD_PROMPT = """
You are an auditor. Count occurrences of the behavior PSD (Pseudo
    implementation) in a competitive-programming reasoning trace.

DEFINITION
PSD = Presenting the algorithm as structured steps or pseudocode with
    control flow, without full code.
Include: numbered/indented outlines; loops/ifs; while/for; state
    updates in an algorithmic outline.

COUNT
- Count 1 per PSD-labeled step.

OUTPUT (strict JSON ONLY):
{
  "PSD": <integer count>,
  "events": [
    {"snippet": "<short quote>", "reason": "<why it matches PSD>"}
  ]
}

<TRACE>
{TRACE}
</TRACE>

Analyze the trace and count the occurrences of PSD.
{TRACE}
"""
```

Prompt 8: PSD_PROMPT (Pseudo Implementation)

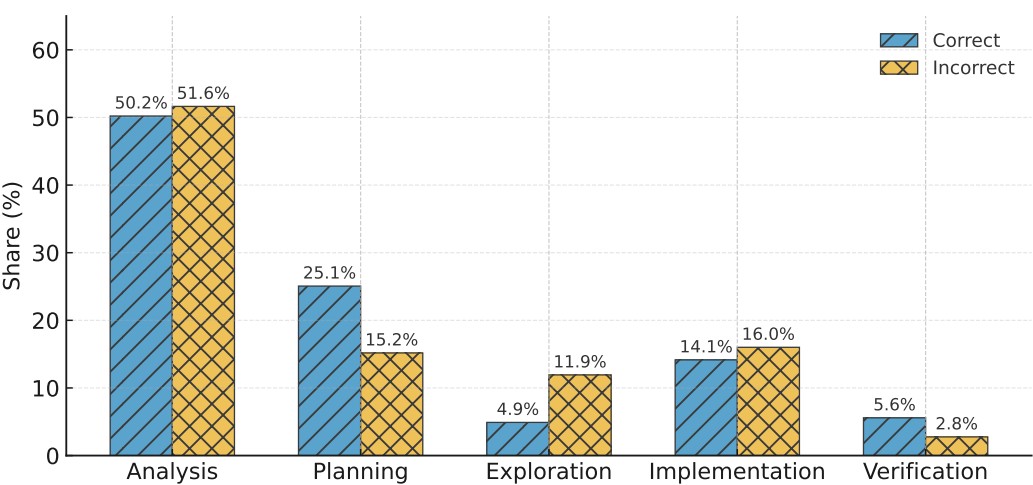

Figure A10: The reasoning behaviors of GPT-OSS-120B-High on easy problems across correct and incorrect solutions. Plan and verification behaviors are still important for models to produce correct solutions.

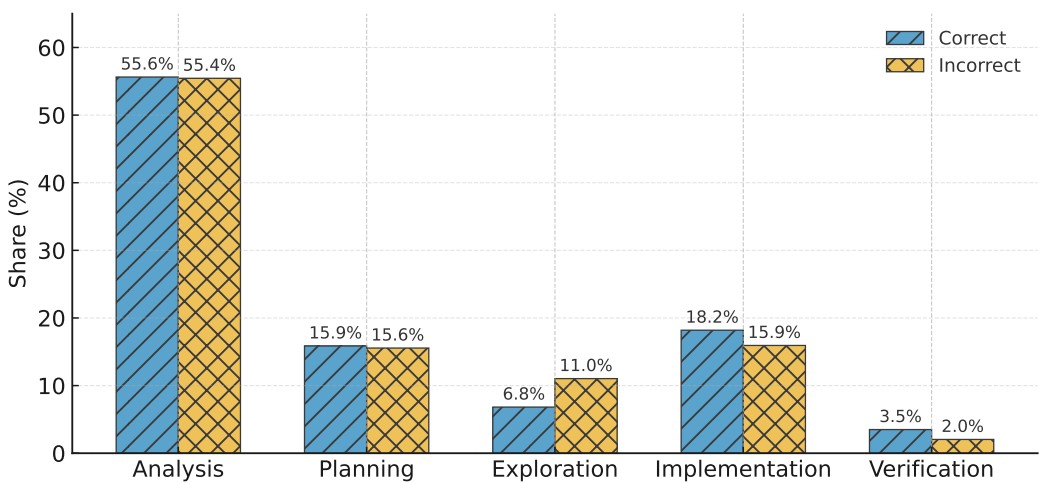

Figure A11: The reasoning behaviors of GPT-OSS-120B-High on medium problems across correct and incorrect solutions. Similar to easy problems, there is less exploration and more verification behaviors for correct solutions.

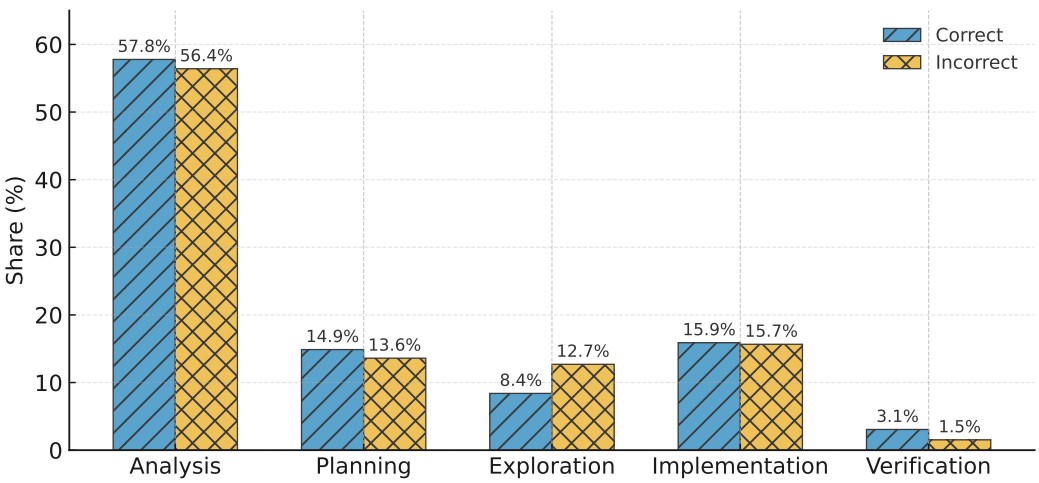

Figure A12: The reasoning behaviors of GPT-OSS-120B-High on hard problems across correct and incorrect solutions. Analysis, plan, and verification behaviors are still important for models to produce correct solutions.

### F.9 REASONING BEHAVIOR ANALYSIS QUALITY CHECK

#### F.9.1 MANUAL ERROR ANALYSIS

We select 10 reasoning traces for GPT-OSS-120B analysis and develop a web visualizer to clearly display the detected behaviors. Based on definitions of each behavior category, we manually inspect all 313 detected behaviors from these 10 reasoning traces and assign correctness labels accordingly. Additionally, we manually identify and annotate any behaviors imissed by GPT-OSS-120B.

Through this careful manual validation, we find that 269 out of the 313 behaviors (86%) are correctly classified. The 44 misclassified behaviors primarily stem from misclassification errors—for instance, GPT-OSS-120B sometimes exhibits oversensitivity to code-like language, misclassifying simple "if-else" analysis statements as pseudo-implementation due to the misleading presence of phrases such as "implementation steps." Another common source of errors includes information loss due to chunking and occasional hallucinations of non-existent behaviors. Additionally, we identify 32 behaviors (approximately 10% relative to detected behaviors) that GPT-OSS-120B fails to detect. These missing behaviors are evenly distributed across all five behavior categories without distinct

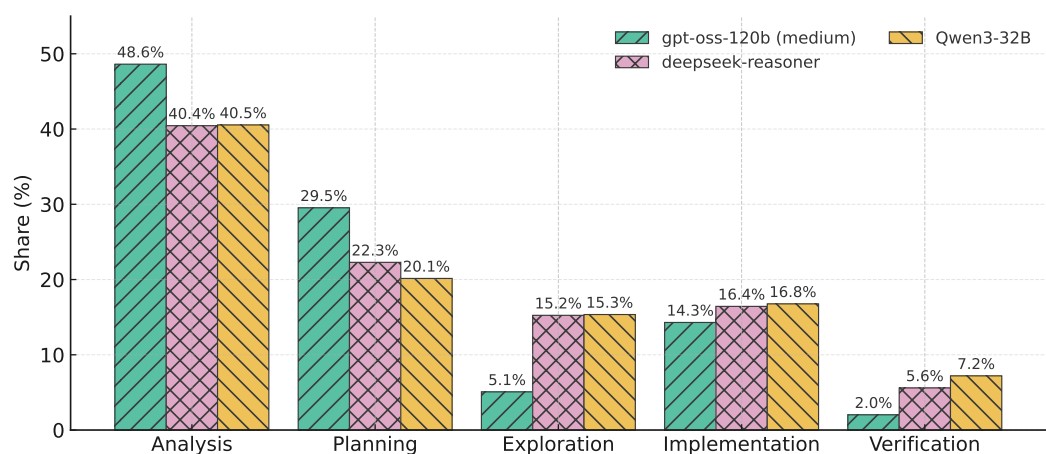

Figure A13: The reasoning behaviors of models producing correct solutions. Stronger reasoning models reduce unnecessary exploration, dedicating more resources to planning, structured analysis, and solution development.

error patterns, consisting of both straightforward and challenging examples. We also show a reasoning trace example in Figure A14. Overall, the manual analysis confirms that GPT-OSS-120B effectively identifies a significant majority of behaviors with robust accuracy, demonstrating substantial reliability in capturing model reasoning behaviors.

### F.9.2 INTER-MODEL AGREEMENT

To further strengthen the validation, we randomly sample 50 additional reasoning traces and pass them into Claude-Sonnet-4.5. Given the complexity and varying granularity of model annotations, instead of directly comparing text spans, we compute Krippendorff's alpha between the two models based on normalized behavior counts. Specifically, we calculate the percentage share of each behavior detected by models and compare their ranking across 50 traces. We observe the following inter-model reliability scores: analysis (=0.8018), planning (=0.7138), exploration (=0.8216), implementation (=0.6895), and verification (=0.6325). The two models show strong and consistent correlation for analysis and exploration behaviors, moderately strong agreement for planning and implementation behaviors, and moderate but slightly lower correlation for verification behaviors. These results further support the reliability and consistency of behavior classifications performed by GPT-OSS-120B.

---

**GPT-OSS-120B-Medium's trace on USACO-2025-OpenContest-Sequence_Construction**

[1] **[Planning]** We need to construct sequence length up to 100 with sum M and xor of popcounts equals K.

[2] ... (7 lines omitted)

[3] **[Planning]** Goal: Find multiset of popcounts ($c_i$) ($0 <= c_i <= 30$) such that xor of $c_i$ = K and sum of some numbers with those popcounts equals M.

[4] ... (1 line omitted)

[5] **[Analysis]** However we can realize any popcount c can be realized by a number with value at most $(2^c - 1)(e.g., lowerbits)$.

[6] ... (86 lines omitted)

[7] **[Planning]** Thus problem reduces to: For given desired popcount c and target value S ($0<=S<=M$), does there exist a non-negative integer x with popcount(x)=c and x=S?

---

[8] ... (46 lines omitted)

[9] **[Analysis]** Thus we can start with binary decomposition of M into bits: we have list L of powers-of-two (size = popcount(M)).

[10] ... (19 lines omitted)

[11] **[Analysis]** If we split a number with popcount c into c ones: before popcount = c, after xor = (1 xor 1 ...

[12] ... (29 lines omitted)

[13] **[Implementation]** Implementation plan: - Decompose M into list of powers-of-two bits (each value = $2^i$).**[INCORRECT]**

[14] ... (28 lines omitted)

[15] **[Verification]** Let's test: starting X parity = (number of bits) %2 = popcount(M) %2.
  M=10 (1010) popcount=2, X=0.
  Target K=5 (101) delta=5 xor 0 =5 (odd).
  Since delta odd cannot be achieved by merging distinct bits (even changes).
  So maybe impossible, matching sample.

[16] ... (9 lines omitted)

[17] **[Verification]** Check other samples: M=2 (10) popcount=1 odd, K=1 odd -> same parity -> possible.

[18] **[Verification]** M=33 (100001) popcount=2 even, K=5 odd => different parity, but sample says possible!

[19] ... (33 lines omitted)

[20] **[Analysis]** If not found with 2 numbers, try 3 numbers: enumerate a,b from set S (size few thousand) and compute c = M-a-b, check if pc(c) = K xor pc(a) xor pc(b).
  Ensure c >=0.
  This is $O(|S|^2)$ ~ few million per test, okay for 5k?

[21] ... (22 lines omitted)

[22] **[Implementation]** Idea: Use DP over bits to construct a with given popcount p such that popcount(M-a)=q. **[INCORRECT]**

[23] **[Analysis]** We can DP over bits with carry (borrow) state.
  Define processing bits from LSB to MSB with borrow flag (0/1).
  For each bit i, we decide $a_i$ (0/1).
  Then $b_i = (M_i - a_i - \text{borrow}) \bmod 2$, and new borrow = $(a_i + \text{borrow} > M_i)$ ? 1 : 0.
  We also count popcounts of a and b.
  DP can compute if reachable.
  We need to know if there exists a with pc(a)=p and pc(b)=q.
  Since p,q <=31, DP state size: position up to 30, borrow 0/1, countA up to 31, countB up to 31.
  That's 31*31*2*31=~60000 manageable.

[24] ... (11 lines omitted)

[25] **[Analysis]** Implement DP bool dp[pos][borrow][pa][pb] reachable.
Initialize dp[0][0][0][0] = true.
Iterate bits i from 0..30 (since $M < 2^{30}$).
At each step, known $M_i$ (0/1).
For $a_i$ in {0,1}.
Then compute diff = $M_i - a_i -$ borrow.
$b_i$ = (diff & 1) (i.e., (diff+2)%2).
newBorrow = diff<0 ? 1 : 0.
Update counts.
At end pos=30, require borrow=0.
Then we have set of reachable (pa,pb).

[26] ... (2 lines omitted)

[27] **[Analysis]** Then we have set of reachable (pa,pb).

[28] **[Analysis]** If there exists pa such that pa xor pb == K, then we have solution with two numbers.

[29] ... (1 line omitted)

[30] **[Analysis]** We can also reconstruct by storing predecessor.

[31] **[Exploration]** If no such pair, we can try three numbers?

[32] **[Exploration]** If no such pair, we can try three numbers?

[33] ... (6 lines omitted)

[34] **[Exploration]** But we can extend DP to three numbers?

[35] ... (8 lines omitted)

[36] **[Exploration]** Simplify: Use DP to find any multiset of up to 5 numbers (since N<=100) with small popcounts by using DP over count of numbers?
Too heavy.
Given time, maybe we can produce algorithm: Use DP to find two numbers (a,b).

[37] ... (1 line omitted)

[38] **[Analysis]** Given time, maybe we can produce algorithm: Use DP to find two numbers (a,b).

[39] ... (12 lines omitted)

[40] **[Exploration]** if not found, output -1. **[INCORRECT]**

[41] ... (29 lines omitted)

[42] **[Analysis]** Complexities: dp size 31*2*32*32  63k bool, fine.

[43] **[Implementation]** Implement predecessor arrays as struct of 5 bits maybe small.
We'll store prev[bit][borrow][pa][pb] with struct containing prevBorrow, prevPa, prevPb, a_bit.
Also b_bit can be derived.
During transition we need b_bit which depends on M_i, a_bit, borrow.
We'll store a_bit only;

> backtrack to compute.
> Implement loops.
> Edge: maximum popcount up to 30, but counts may go up to 31 (if all bits set).
> Use 32 size.
> Implementation details: int MAXB=31;
> // bits 0..30 for (int i=0;i<MAXB;i++) { // i from 0 LSB int mi = (M»i)&1;
> for borrow...
> for pa...
> for pb...
> if dp[i][borrow][pa][pb] true for a_bit in {0,1} int diff = mi - a_bit - borrow;
> ... (29 lines omitted)

Figure A14: Behavior-labeled reasoning trace for the *Sequence Construction* problem (grouped behavior types shown in-line). We map low-level labels into five categories: **Planning** (PR/SUB), **Analysis** (AP/CMP), **Exploration** (BKT/DED), **Implementation** (PSD), and **Verification** (V–T). Misclassified events are marked with **[INCORRECT]**; across our evaluation on 10 reasoing traces, the 44 misclassifications primarily arise from (i) oversensitivity to code-like phrasing (e.g., "implementation steps" triggering pseudo-implementation), (ii) information loss due to chunking, and (iii) occasional hallucinated (non-existent) behaviors.

