# OpenReview forum: "LiveOIBench: Can Large Language Models Outperform Human Contestants in Informatics Olympiads?"
_ICLR.cc/2026/Conference — Submitted to ICLR 2026_

### Official Review · Reviewer_RVy3 · 2025-10-29

**Soundness:** 2
**Presentation:** 4
**Contribution:** 3
**Rating:** 6
**Confidence:** 3

**Summary:**

This paper introduces LiveOIBench, a comprehensive benchmark designed to evaluate the coding and algorithmic reasoning capabilities of large language models (LLMs). The benchmark addresses several critical limitations of existing coding datasets, such as test case insufficiency, lack of exceptionally difficult problems, and reliance on external APIs that hinder reproducibility. Furthermore, the benchmark integrates official human contestant data, enabling direct comparisons through metrics like human percentiles and ELO ratings. The authors' evaluation of 32 models reveals that even top performers like GPT-5, while achieving an impressive 81st percentile, still lag behind elite human contestants, highlighting substantial room for progress in LLM reasoning capabilities.

**Strengths:**

* The paper presents a much-needed, high-quality benchmark for evaluating the performance of frontier large language models (LLMs) on competitive programming–style (Informatics Olympiad) problems, which demand deep algorithmic reasoning and structured problem solving.
* While prior works such as OI-Bench and LiveCodeBench have explored similar domains, the paper clearly acknowledges these efforts and articulates how LiveOIBench advances beyond them through improved data curation, richer evaluation metrics, and stronger reproducibility.
* A key distinguishing feature is that LiveOIBench is “live” — it continuously incorporates newly released Olympiad contests, ensuring ongoing relevance and contamination-free updates. Moreover, it uniquely benchmarks LLMs directly against human contestant performance, offering an interpretable and intuitive comparison framework
* The breadth and diversity of the dataset — spanning 403 problems from 72 contests across 14 Olympiads — significantly enhances the benchmark’s robustness and difficulty spectrum, making it a valuable resource for the community.
* The evaluation is comprehensive, covering 32 leading proprietary and open-weight models, and complemented by detailed analyses of algorithmic categories (e.g., dynamic programming, graph theory) and reasoning behavior.

**Weaknesses:**

While this is an excellent paper with a significant contribution, there are several areas where it could be strengthened to improve its clarity, reproducibility, and the impact of its conclusions. Listed a few below:

* While the paper includes ablations across different Pass@k values and reasoning budgets, it does not examine the sensitivity of results to prompt structure or instruction phrasing. Given the well-known prompt sensitivity of LLMs, an analysis using alternative templates (e.g., explicit “think step-by-step” or language-specific wrappers) would strengthen the conclusions about model robustness and fairness under different interaction styles.
* Will be good if authors can add generation parameters such as temperature and top_p in the draft. Explicitly listing these values (even in the appendix) would enhance transparency.
* While the paper mentions that the “full problem statement is presented to the models,” it does not provide the exact prompt wrapper or instruction format used to elicit code generation (e.g., whether the prompt included directives like “write code in C++ that reads from stdin”). Adding this will enhance the readability.
* The paper does not explicitly mention which programming language the models were instructed to generate code in. I am assuming the outputs were in C++ (correct me if I am wrong), since Olympiads such as IOI and JOI traditionally restrict submissions to that language. However, it would be interesting to see whether the models were allowed to generate solutions in other languages such as Python or Java, as contests like USACO permit multiple languages. Add this detail will help.
* Also, evaluating cross-language performance could provide valuable insight into the models’ adaptability and coding generalization abilities. Authors may consider adding this in the draft.
* The reasoning-trace study (Sec. 5.2) classifies model behavior into five categories using GPT-OSS-120B itself, but there is no error analysis or validation performed on how accurately GPT-OSS-120B-High classifies the reasoning traces into these buckets. Authors may consider adding this in the draft. While it would be more robust to involve multiple LLMs for this classification task and compute an inter-annotator agreement metric (such as Cohen’s κ) to assess consistency, even a smaller-scale manual validation or qualitative error analysis would strengthen the credibility of the findings. For instance, the paper could include an example reasoning trace generated by one of the LLMs on a particular problem and then illustrate how GPT-OSS-120B-High assigns it to the five reasoning categories, making the analysis more transparent and interpretable.

**Questions:**

I encourage the authors to address the specific weaknesses raised above, as clarifications or additional evidence on these points could substantially strengthen the paper. In particular, I would appreciate responses or additional details on:

1. What programming languages were allowed, and what prompt template was used (include the wrapper text such as ‘write C++ code that reads from stdin’)?

2. What were the decoding hyper parameters used during evaluation?

3. Clarify the update window and whether a hidden, rotating holdout will be kept for future releases?

4. Will be great if authors can conduct qualitative analyses with multiple prompt variations to evaluate the stability and robustness of model performance under different prompting conditions, and report the results?

---

> ### Author Response · Authors · 2025-11-24
> **Response to Reviewer RVy3 (1/3)**
>
> Thank you for recognizing our work contribution and insightful comments. We aim to address your questions in the following:
>
> **W1 + Q4**
> In Appendix F.6, we generate four alternative prompt variants using GPT-5 and run GPT-OSS-120B-Medium with four additional prompt variants to explore how different instructions might influence the model’s performance. While these prompts do not represent all possible instruction styles, we observe minimal performance differences across them, suggesting that prompt variants have little influence on model performance. A more systematic and large-scale study of prompt sensitivity is an interesting direction for future work. The default prompt is shown in our response to W3.
>
> **Prompt 1:**
>
> You are a helpful and knowledgeable AI assistant. Answer the user's questions clearly and concisely, using correct reasoning and accurate information.
>
> **Prompt 2:**
>
> You are an expert IOI competitive programming assistant. For each problem, output only a single complete C++17 solution that compiles and solves the task. Do not include explanations, comments, or any text outside one ```cpp``` code block. Read input from standard input and write output to standard output.
>
> **Prompt 3:**
>
> You are an expert IOI problem solver. First, think through the solution step by step and explain your reasoning in a few clear paragraphs or bullet points. After that, output a single complete C++17 solution inside one ```cpp``` code block that follows your reasoning, reads from standard input, and writes to standard output.
>
> **Prompt 4:**
>
> You are a meticulous and safe IOI problem-solving assistant. Always prioritize correctness, handling of corner cases, and numerical stability over constant-factor performance. Carefully verify your reasoning before producing an answer. When writing code, output a single complete C++17 solution in a ```cpp``` block, making sure it compiles, handles invalid or extreme inputs robustly when appropriate, and does not perform unsafe or undefined operations.
>
> | Model               | Relative Score (%) | Pass Rate (%) | Human Percentile |
> |---------------------|--------------------|----------------|------------------|
> | gpt-oss-120b_p1    | 47.76              | 45.85          | 58.5             |
> | gpt-oss-120b_p2    | 48.07              | 47.10          | 59.76            |
> | gpt-oss-120b_p3    | 46.75              | 45.69          | 57.73            |
> | gpt-oss-120b_p4    | 48.04              | 45.68          | 59.49            |
> | gpt-oss-120b-medium | 48.88              | 47.33          | 58.1             |
> | **Mean ± Std (all five)**     | **47.90 ± 0.69**   | **46.33 ± 0.73** | **58.72 ± 0.79** |
>
> Across all five prompt variants, GPT-OSS-120B achieves a relative score of 47.9 ± 0.7%, a pass rate of 46.3 ± 0.7%, and a human-percentile score of 58.7 ± 0.8, indicating low sensitivity to sampling choices.

---

> ### Author Response · Authors · 2025-11-24
> **Response to Reviewer RVy3 (2/3)**
>
> W2 + Q2: generation parameter
>
> In the updated PDF, we have included the generation parameters of all 34 models. Please see Table A7 and Table A8. For the models in our main result table, here are the decoding parameters we use.
>
> | **Models**                      | **Model Provider** | **Reasoning Budget (Max Tokens)** | **Temperature** | **Top-p** |
> |----------------------------------------|--------------------|-----------------------------------|-----------------|-----------|
> | GPT-5                                  | OpenAI             | Medium (100K)                     | -               | -         |
> | GPT-O3-Mini-High                       | OpenAI             | High (100K)                       | -               | -         |
> | GPT-OSS-120B                           | OpenAI             | Medium (128K)                     | 1.0             | 1.0       |
> | GPT-OSS-20B                            | OpenAI             | Medium (128K)                     | 1.0             | 1.0       |
> | Grok-4-Fast-Reasoning                  | xAI                | 100K                              | 1.0             | 1.0       |
> | Claude-Sonnet-4.5                      | Anthropic          | 120K                              | 1.0             | 1.0       |
> | SEED-OSS                               | ByteDance          | Unlimited (128K)                  | 1.1             | 0.95      |
> | Qwen3-32B                              | Alibaba            | 38K                               | 0.6             | 0.95      |
> | Qwen3-14B                              | Alibaba            | 38K                               | 0.6             | 0.95      |
> | DeepSeek-R1-01-28                      | DeepSeek           | 32K                               | 0.6             | 0.95      |
> | Gemini-2.5-Pro-exp-03-25               | Google             | 64K                               | 1.0             | 0.95      |
> | Gemini-2.5-Flash-preview-04-17         | Google             | 64K                               | 1.0             | 0.95      |
> | DeepSeek-R1-Distill-Llama-70B          | DeepSeek           | 32K                               | 0.6             | 0.95      |
> | GPT-4.1                                | OpenAI             | -                                 | 1.0             | 1.0       |
> | DeepSeek-V3                            | DeepSeek           | -                                 | 1.0             | 1.0       |

---

> ### Author Response · Authors · 2025-11-24
> **Response to Reviewer RVy3 (3/3)**
>
> **W3** -> Below are the instructions we use to evaluate the models, and we also include this prompt template in our appendix.
>
> **Prompt:**
> Given a competition problem below, write a solution in C++ that solves all the subtasks. Make sure to wrap your code in '```<task>.cpp' and '```' Markdown delimiters.
>
> [BEGIN PROBLEM]
>
> [task statement]
>
> [END PROBLEM]
>
> Time limit: [time] seconds
>
> Memory limit: [memory] mb
>
> Generate a solution in C++ that solves the task. Make sure to wrap your code in '```<task>.cpp' and '```' Markdown delimiters.
>
> **W4 + W5** -> All evaluations in our benchmark are conducted using C++, as it is the official language for most competitive programming contests and is widely favored in the competitive coding community for its efficiency and performance advantages. We present a comparative analysis (Appendix F.3) of GPT-OSS-120B’s performance across Python, Java, and C++, evaluated on a subset of 132 problems from the USACO contest series. We find that C++'s performance advantage mainly stems from its computational efficiency rather than variations in reasoning approach. Please read this comment: Performance variation across programming languages (https://openreview.net/forum?id=URtz3JhoWh&noteId=evwVSNYZLK)
>
> **W6** -> Thank you for highlighting this issue. We're currently performing an error analysis on GPT-OSS-120B traces. While the analysis is still underway, we will provide detailed case-study examples as soon as possible.
>
> **Q3** -> We plan to release a new version of our datasets every 3-6 months and will not keep the hidden set since all the informatics olympiads are available online to everyone.

---

> ### Author Response · Authors · 2025-12-03
>
> **W6** → Given the time constraints and extensive length of the reasoning traces, we randomly selected 10 reasoning traces from GPT-OSS-120B and manually inspected all 313 detected behaviors (approximately 31 behaviors per trace). Through this careful manual validation, we found that 269 out of the 313 behaviors (**86% accuracy**) were correctly classified, while also identifying 32 behaviors (10% relative to the total behaviors) that GPT-OSS-120B missed. Further details regarding the manual error analysis can be found in Appendix F.9.1. An example reasoning trace can be found in Figure A14. Overall, the manual analysis confirms that GPT-OSS-120B **accurately** identifies the significant majority of behaviors, demonstrating substantial reliability in capturing model reasoning behaviors.
>
> To further strengthen this validation, we randomly sampled an additional 50 reasoning traces and passed them into Claude-Sonnet-4.5. We computed Krippendorff's alpha between the two models and observed **an average inter-model agreement of 0.7314** across five categories, indicating **consistent** agreement between the two models. More details on this inter-model agreement calculation can also be found in Appendix F.9.2. This result further supports the reliability and consistency of the behavior classifications performed by GPT-OSS-120B.

---

### Official Review · Reviewer_faYZ · 2025-10-30

**Soundness:** 3
**Presentation:** 3
**Contribution:** 3
**Rating:** 6
**Confidence:** 3

**Summary:**

This paper introduces LiveOIBench, a competitive programming benchmark derived from Informatics Olympiad contests. It comprises 403 problems with detailed subtask rubrics and expert curated private test cases, supporting offline evaluation. The authors benchmark 32 models and find that GPT-5 achieves the best average performance yet still exhibits a marginal gap compared to top-tier human contestants. They also observe that thinking models outperform non-thinking variants and that higher-performing models allocate more tokens to analysis rather than implementation.

**Strengths:**

- The paper accurately identifies limitations of current code benchmarks and constructs a high-quality benchmark that addresses those issues.

- LiveOIBench contains 400+ competitive problems, which is large-scale and high-difficulty relative to existing datasets and can provide a valuable evaluation framework for future works.

- The paper conducts substantial experiments and surfaces several clear empirical insights.

**Weaknesses:**

- Because all problems come from real OI contests, newer models may have been trained on these problems and their solutions, which could compromise the reliability of the experimental results due to potential data contamination.

- There may be mismatches or missing mappings between contestants and their Codeforces profiles, which could affect the accuracy of Codeforces Elo computations and percentile estimates.

**Questions:**

1. The paper states (around line 200) that 40 tasks were sampled to check PDF-to-Markdown conversion accuracy. What were the results, and how do you ensure high conversion quality for the remaining tasks?

2. The paper does not specify the programming languages used during evaluation. Since contestants may use a range of languages, what is the impact of language choice on the reported metrics?

---

> ### Author Response · Authors · 2025-11-24
> **Response to Reviewer faYZ (1/2)**
>
> Thank you for recognizing our work contribution and insightful comments. We aim to address your questions in the following:
>
> **W1** -> We have added additional analyses on both data contamination and template-based contamination in Section 5.4. Based on these results, we believe contamination in our benchmark is minimal. We will also continue updating the benchmark with newly released contests every 3–6 months. Please refer to this comment: Additional analysis on data contamination (https://openreview.net/forum?id=URtz3JhoWh&noteId=LXVmLvmL3B).
>
> **W2** -> To calculate the percentile of each model, we collect the official ranking from each contest and compare the model’s score with those of other contestants. The human percentile estimates will be **accurate** since we directly compare the model against other humans in the contests.
>
> To ensure accurate mapping between contestants and their Codeforces profiles, our collection pipeline employs several heuristics to filter out irrelevant or low-quality matches. Initially, we construct a search query using each contestant’s name and country, appended with "Codeforces," and utilize the Google Search API to retrieve relevant profile URLs. Once valid profiles are fetched, their relevance is assessed by comparing each contestant’s name and country with the corresponding details on the profile, discarding profiles exhibiting less than 50% similarity.
>
> For profiles showing moderate similarity (50%-70%), we employ Gemini-2.0, enhanced with search capabilities, to make a definitive relevance determination. For contests like JOI, where contestants typically use their real names as usernames, we directly access their Codeforces rating history.
>
> After gathering profile data, contestants without ratings or those with ratings below 500 are excluded. We then perform a linear regression analysis of score-rating data within each contest, calculating the residuals' standard deviation. Contestants whose ratings deviate by more than two standard deviations from the regression line—suggesting a potential mismatch—are removed as outliers. Finally, we exclude any contests from analysis that contain fewer than 15 verified contestant profiles to maintain robust and reliable rating computations.
>
> To verify the accuracy of matching contestants to their Codeforces profiles, we performed **a manual error analysis**. We randomly sampled five contest results, selecting ten contestants from each contest. Using the same search query (contestant name, country, and "Codeforces"), we located each contestant's Codeforces profile and verified the match by comparing the name, country, username, and yearly rating history against our stored records.. Our analysis confirmed that all 50 contestant profiles matched perfectly, demonstrating 100% accuracy and confirming the **high precision** of our profile-matching pipeline. Nevertheless, we recognize that some contestants may lack Codeforces profiles or use alternative names on their profiles, potentially causing missed matches. Given the high precision of our pipeline, the individualized nature of ELO rating computations, and our exclusion of contests containing fewer than 15 validated contestant profiles, we anticipate that such omissions will **minimally** affect the overall accuracy of the model’s rating estimations.

---

> ### Author Response · Authors · 2025-11-24
> **Response to Reviewer faYZ (2/2)**
>
> **Q1** -> We sampled 40 tasks (around 10% of task statements) maintaining the same distribution across competition types as the full dataset. Our analysis revealed 4 tasks (10%) with conversion errors, primarily in nested table parsing within example inputs/outputs. While such formatting inconsistencies might impact human understanding, they have **minimal effect** on LLM comprehension as the core question semantic remains intact. GPT-5 fully solved three of the four tasks and passed over 70% of test cases on the remaining one, demonstrating that models are less sensitive to these formatting issues. An additional 14 tasks (35%) exhibited minor formatting issues, including header-level inconsistencies (e.g., "Input" formatted as # Input while "Output" is formatted as ### Output despite being at the same hierarchical level), incorrect superscript rendering, and uppercase/lowercase mismatches. These are inconsequential formatting artifacts that do not alter the semantic meaning of the problems. Our conversion pipeline uses Marker[1] for PDF-to-markdown conversion followed by Gemini-2.0-Flash for automated verification and correction. After manually reviewing this 40-task sample, we confirmed that the parsed task statements are indeed of **high quality**, with only minor formatting errors that do not impact model performance. Confident in the robustness of these results, we proceeded with batch processing the remaining tasks. We believe these slight formatting inconsistencies are inconsequential and do not compromise the validity of our evaluation.
>
> **Q2** -> All evaluations in our benchmark are conducted using C++, as it is the official language for most competitive programming contests and is widely favored in the competitive coding community for its efficiency and performance advantages. We present a comparative analysis (Appendix F.3) of GPT-OSS-120B’s performance across Python, Java, and C++, evaluated on a subset of 132 problems from the USACO contest series. We find that C++'s performance advantage mainly stems from its computational efficiency rather than variations in reasoning approach. Please read this comment: Performance variation across programming languages (https://openreview.net/forum?id=URtz3JhoWh&noteId=evwVSNYZLK)
>
> [1] Marker. https://github.com/datalab-to/marker.

---

### Official Review · Reviewer_rxTf · 2025-11-03

**Soundness:** 4
**Presentation:** 3
**Contribution:** 3
**Rating:** 6
**Confidence:** 4

**Summary:**

The paper pulls together a high quality dataset to test the reasoning ability of frontier LLMs on difficult coding problems and compares their performance to top human reasoners. It does extensive work to setup a website that will pull new challenging problems in from future coding contests so that a fresher dataset is always available to counter the contamination that happens as older test sets and their solutions spread on the internet.

**Strengths:**

A very challenging dataset of coding infomatic style problems is made, with a nice website with a leaderboard to compare various LLMs performance. The dataset is contamination resistant by being designed to be updated with new problems from future contests.

**Weaknesses:**

I would have liked to see more baselines for other model providers like Anthropic or X.AI

**Questions:**

Can you add baselines for the other major closed models?  I think the performance of more frontier models on the dataset would make the paper more valuable.

---

> ### Author Response · Authors · 2025-11-24
> **Response to Reviewer rxTf  (1/1)**
>
> Thank you for your insightful comments and feedback. We have added additional results on Claude Sonnet-4.5 and Grok-4-Fast-Reasoning in Table 2. We also attach the result table below.
>
> | Model                                | Gold (%) | Medals (%) | Relative Score (%) | Human Percentile (%) | Pass Rate (%) | ELO  |
> |--------------------------------------|----------|------------|---------------------|------------------------|----------------|------|
> | GPT-5                                | **50.00** | **88.89** | **67.21**           | **81.76**              | **63.03**      | **2414** |
> | Grok-4-Fast-Reasoning | 45.83   | 83.33     | 56.99                | 74.23                 | 50.95         | 2221|
> | Gemini-2.5-Pro                       | _31.94_  | 77.78      | 51.33                | 71.80                 | 44.46         | 2192 |
> | GPT-O3-Mini-High                     | 26.39    | 72.22      | 47.69                | 64.28                 | 44.19         | 2088 |
> | Gemini-2.5-Flash                     | 15.28    | 62.50      | 41.29                | 56.81                 | 36.06         | 1945 |
> | Claude-Sonnet-4.5    | 11.11    | 66.68      | 38.30                | 53.08                 | 27.05         | 1848 |
> | GPT-4.1                              | 4.17     | 40.28      | 24.78                | 35.99                 | 18.32         | 1482 |
>
> Grok-4-Fast-Reasoning shows very competitive performance, with a human percentile of 74.23, ranking second-highest among all models. Claude-Sonnet-4.5 lags behind all other thinking proprietary models, ranking only in the 56.81th percentile. The poor performance of Claude-Sonnet-4.5 is also observed in other coding benchmarks, such as LiveCodeBench Pro.

---

### Official Review · Reviewer_Xz3c · 2025-11-03

**Soundness:** 2
**Presentation:** 2
**Contribution:** 2
**Rating:** 4
**Confidence:** 4

**Summary:**

This paper introduces LiveOIBench, a benchmark of 403 Olympiad-level coding problems from 72 contests to evaluate LLMs’ algorithmic reasoning. It features expert-written test cases, subtask rubrics, and human contestant data. Evaluations of 32 models show GPT-5 reaches the 82nd human percentile, still below top human coders. The benchmark is comprehensive, reproducible, and contamination-free, offering a new standard for assessing LLM coding ability.

**Strengths:**

1. Expert-curated Olympiad problems with rich subtasks and reliable private tests.
2. Direct comparison to human contestants enables meaningful percentile evaluation.
3. Comprehensive, reproducible benchmark revealing detailed reasoning and algorithmic insights.

**Weaknesses:**

1. The analysis and conclusion are not deep, providing limited new insights beyond benchmark construction.
2. The paper mainly focuses on dataset creation and evaluation, making it more suitable for a dataset or benchmark track rather than the main ICLR track.

**Questions:**

1. How do you ensure the fairness and consistency of problem difficulty and judging standards across the 14 Olympiads (e.g., NOI vs. ICPC-style tasks), given their distinct styles and scoring rubrics?
2. Could you elaborate on why C++ appears to yield better outcomes in your experiments—do models genuinely reason differently in C++, or is this advantage primarily due to compilation or runtime efficiency effects?
3. Have you analyzed whether model size, reasoning token allocation, or RL/distillation strategies correlate more strongly with success on algorithmic tasks like dynamic programming?
4. Given the benchmark’s reliance on Olympiad data, what measures are in place to detect or mitigate data leakage, especially since some contest problems or close variants might exist in public training corpora used by large models?

---

> ### Author Response · Authors · 2025-11-24
> **Response to Reviewer Xz3c (1/2)**
>
> Thank you for recognizing our work contribution and insightful comments. We aim to address your questions in the following:
>
> **Weakness 1** -> In the revised PDF, we've expanded our analyses to address your concerns. In addition to our initial findings, we now include deeper analyses on **data contamination** (Section 5.4; demonstrating minimal influence from solution-template similarity), **performance variation across programming languages** (Appendix F.3; confirming superior performance in C++ due to efficiency), **algorithmic tags** (Appendix F.5; showing reasoning-token budgets significantly enhance performance on complex tasks), and **prompt sensitivity** (Appendix F.6; highlighting GPT-OSS-120B's low sensitivity to instructions). If you have specific suggestions or additional directions for meaningful analyses, please let us know—we would be glad to incorporate your feedback to strengthen the paper further.
>
> **Weakness 2** -> We appreciate your suggestion. However, we would like to clarify that ICLR does not offer a separate track specifically dedicated to datasets and benchmarks. Accordingly, we intentionally selected "Datasets and Benchmarks" as our primary submission area, which aligns precisely with the core contributions of our paper.
>
> **Q1** -> Our benchmark only includes informatics olympiad tasks to maintain consistency in task structure and evaluation methods. To ensure fairness and standardization, we source official test cases and detailed scoring rubrics directly from each Olympiad's official materials. These test cases and rubrics are then rigorously validated against the provided official solutions.
> Additionally, we gather contestant rankings from the official Olympiad websites and systematically annotate problem difficulty levels using reputable online judging platforms, such as Luogu and solved.ac. Difficulty tags from these platforms are normalized onto a unified 1–30 scale for consistency across contests.
> Recognizing inherent variability in problem difficulty across the 14 different Olympiads, we further categorize our problems into three clearly defined levels—Easy, Medium, and Hard—and organize contests into four distinct divisions based on their overall difficulty. The distributions of problems and contests are detailed in Table A2 and Table A3 and are also provided below. Furthermore, model performance across problem difficulty levels and contest divisions is provided in Table A9.
> | **Level** | **# Problems** | **% of Total** | **Threshold Rule**     |
> |-----------|----------------|----------------|-------------------------|
> | Easy      | 143            | 35.48%         | \( d <=17 \)          |
> | Medium    | 144            | 35.73%         | \( 18 <= d <= 22 \)   |
> | Hard      | 116            | 28.78%         | \( d >= 23 \)          |
> | **Total** | **403**        | **100%**       | --                      |
>
> | **Division** | **Min Difficulty** | **Max Difficulty** | **Avg Difficulty** | **Total Contests** |
> |--------------|--------------------|--------------------|--------------------|---------------------|
> | Division 4   | 5.0                | 15.78              | 13.76              | 17                  |
> | Division 3   | 16.0               | 20.33              | 18.05              | 19                  |
> | Division 2   | 20.33              | 22.33              | 21.52              | 19                  |
> | Division 1   | 22.5               | 30.0               | 23.62              | 17                  |
>
> **Q2** -> All evaluations in our benchmark are conducted using C++, as it is the official language for most competitive programming contests and is widely favored in the competitive coding community for its efficiency and performance advantages. We present a comparative analysis (Appendix F.3) of GPT-OSS-120B’s performance across Python, Java, and C++, evaluated on a subset of 132 problems from the USACO contest series. We find that C++'s performance advantage stems from its computational efficiency rather than variations in reasoning approach. Please read this comment: Performance variation across programming languages (https://openreview.net/forum?id=URtz3JhoWh&noteId=evwVSNYZLK)

---

> ### Author Response · Authors · 2025-11-24
> **Response to Reviewer Xz3c (2/2)**
>
> **Q3** ->
> In Appendix F.5 and Table A12, we present a detailed analysis of model performance across various algorithmic categories, examining three primary factors: model size, reasoning-token budget, and training strategies (RL and distillation).
>
> **Model Size:** Scaling GPT-OSS from 20B to 120B parameters under low-reasoning effort yields an average improvement of +18%. This gain is concentrated mainly in Greedy (+83.30%), Graph Traversal (+42.80%), Combinatorics (+33.40%), and Tree (+33.40%). These patterns indicate that additional parameters enhance the model’s structural representations and pattern-recognition capacity, enabling better handling of connectivity and global relational structure.
>
> **Reasoning Tokens:** Increasing the reasoning-token budget from low to high, while maintaining the same GPT-OSS-20B model weights, produces a substantially larger average improvement of +113%. Gains are most pronounced in algorithmic tasks such as string manipulation (+268.33%), tree traversal (+220.00%), greedy algorithms (+212.64%), dynamic programming (+186.43%), and sorting (+121.44%),  all of which demand extended multi-step, sequential reasoning. Notably, greedy and tree traversal appear in the top-5 for both factors: larger models recognize their structural patterns more reliably, while deeper reasoning is essential to execute the multi-step, stateful chains these problems demand.
>
> **RL vs. Distillation:** Since each model family differs in architecture, pretraining data, and training pipelines, it’s difficult to cleanly isolate the effects of RL versus distillation on downstream performance. However, preliminary comparisons between Qwen3-14B (a distilled model) and GPT-OSS-20B (trained via SFT and RL) suggest RL-trained models, with SFT warmup and enough RL training, can substantially outperform distilled models, with an average improvement of +118%. We acknowledge that isolating the contribution of each training stage is an important future direction for us to understand how these methods shape algorithmic skills.
>
> Overall, our analyses show that reasoning depth, far more than parameter count, drives improvements on the most algorithmically complex tasks. Also, directly trained RL models perform better than distilled models.
>
> **Q4** ->
> We have added additional analyses on both data contamination and template-based contamination in Section 5.4. Based on these results, we believe contamination in our benchmark is minimal. We will also continue updating the benchmark with newly released contests every 3–6 months. Please refer to this comment: Additional analysis on data contamination (https://openreview.net/forum?id=URtz3JhoWh&noteId=LXVmLvmL3B).

---

### Official Review · Reviewer_bPmJ · 2025-11-08

**Soundness:** 3
**Presentation:** 3
**Contribution:** 2
**Rating:** 4
**Confidence:** 4

**Summary:**

### Paper Summary
This paper presents LiveOIBench, a competitive coding benchmark designed to evaluate large language models (LLMs)’ algorithmic reasoning and programming capabilities. It curates 403 high-quality tasks from 72 official Informatics Olympiad competitions (2023–2025), complemented by abundant official test cases. The benchmark evaluates 32 LLMs across closed-source/open-source and thinking-enabled/non-thinking-enabled categories, using human-centric metrics to compare models against human competitors.

### Core Contributions
1. Constructs a large-scale, reproducible benchmark leveraging recent, authoritative Olympiad tasks, addressing limitations of synthetic or outdated coding benchmarks.
2. Introduces multi-dimensional metrics (e.g., Human Percentile, Codeforces ELO) for intuitive human-model performance comparisons, advancing beyond simplistic pass/fail indicators.
3. Provides granular insights into model strengths/weaknesses via subtask breakdowns and error mode analysis (e.g., compilation vs. logic errors).
4. Validates performance hierarchies across model types, highlighting the value of structured thinking for complex algorithmic tasks.

**Strengths:**

* **Subtasks with Scoring Rubrics** By leveraging the "subtask breakdown" and "hierarchical scoring rubrics" inherent in informatics olympiad problems, the paper breaks down the model's "overall accuracy rate" into "score performance across various ability modules." This enables the acquisition of nuanced insights into "which specific types of tasks the model excels at and which it struggles with," rather than merely providing a vague conclusion about "the model's performance level."
* **Continuous Updates.** The Continuous Updates of LiveOIBench are achieved through five core mechanisms: dynamic task injection, user feedback-driven iteration, versioned management, scoring rule calibration, and delayed release strategy. Its primary goals are to maintain the benchmark’s timeliness, discriminative power, and fairness, while combating data contamination and model overfitting. This design paradigm aligns closely with cutting-edge dynamic benchmarks such as LiveBench and LiveCodeBench, reflecting the latest trends in the current AI evaluation field.

**Weaknesses:**

* **Overclaim** The authors claim that LiveOIBench is `the first comprehensive competitive coding benchmark constructed directly from Informatics Olympiads tasks`—a claim that is not justified given the existence of concurrent work.
* **Insufficient Justification** This paper does not do enough to mitigate risks of data contamination, as template-based contamination has not been assessed to a reasonable extent.

**Questions:**

Whether large language models (LLMs) reason based on memorization remains highly controversial. Most benchmarks attempt to address contamination issues from diverse perspectives. Unlike mathematical reasoning, the reviewer argues that informatics olympiad tasks may share similar algorithmic templates while having different surface forms. Is a Serendipity Metric—designed to calibrate template similarities—necessary to make the benchmark more contamination-free?

---

> ### Author Response · Authors · 2025-11-24
> **Response to Reviewer bPmJ (1/2)**
>
> Thank you for recognizing our work contribution and insightful comments. We aim to address your questions in the following:
>
> **W1 - Overclaim** -> In the updated PDF, we have clarified and expanded upon our original claim to better highlight the unique contributions of LiveOIBench. Specifically, we emphasize that LiveOIBench is the first comprehensive **informatics olympiad** coding benchmark constructed **directly** from official contest sources, integrating **full test cases**, **detailed subtask rubrics**, and **official contestant rankings**. This distinction clearly separates our work from concurrent benchmarks.
>
> LiveCodeBench Pro and HLCE primarily focus on ICPC/IOI problems and evaluate models using Codeforces, which inherently limits their accessibility. OI-Bench, on the other hand, mainly depends on private, non-English-language school competitions and lacks regular updates or maintenance. Similarly, AetherCode relies on LLM-generated tests accompanied by extensive human annotations and only evaluates models using pass rates, not providing comprehensive subtask rubrics or official rankings. In contrast, LiveOIBench uniquely incorporates official hidden test cases, subtask scoring rubrics, and contestant rankings sourced directly from official Informatics Olympiad websites. Our benchmark addresses all these shortcomings by providing full test cases and subtasks for local evaluation, ensuring regular updates synchronized with ongoing Informatics Olympiads, and enabling direct comparisons between model performance and human contestants.
>
> **W2 - Insufficient Justification** -> We have added additional analyses on both data contamination and template-based contamination in Section 5.4. Based on these results, we believe contamination in our benchmark is minimal. We will also continue updating the benchmark with newly released contests every 3–6 months. Please refer to this comment: Additional analysis on data contamination (https://openreview.net/forum?id=URtz3JhoWh&noteId=LXVmLvmL3B).
>
> **Q1 - Is a Serendipity Metric—designed to calibrate template similarities—necessary to make the benchmark more contamination-free?** ->  We believe that Olympiad coding problems are inherently challenging by design, intended explicitly to distinguish among the world’s most skilled competitive programmers. Each problem crafted by the Informatics Olympiad committee is carefully structured to encourage novel, insightful, and creative solutions. Merely memorizing past solutions or algorithmic templates provides limited advantage, as full problem-solving success typically requires genuine algorithmic understanding and creative thinking.
>
> This view is reinforced by our similarity analysis between models' generated solutions and official solutions: the consistently low similarity scores clearly demonstrate that the models are generating original approaches rather than relying on memorized templates. This can be further supported by the template similarity analysis, which shows a negligible correlation between template similarity and model performance differences.
>
> We are conducting further experiments to validate the necessity of a Serendipity Metric by comparing official solutions to templates from a code training corpus; this analysis involves extensive comparisons among many solutions in the code training corpus, requiring more time; we will update our results shortly and welcome any suggestions.
>
> At the end, we would like to emphasize our contributions:
> - **Data.** Curate and release a comprehensive, high-quality competitive coding benchmark with expert-crafted problems, hidden test suites, and integrated human contestant results.
>
> - **Evaluation.** Provide a robust local evaluation framework with private test cases and detailed subtask scoring rubrics, enabling accessible, fine-grained human–model comparisons.
>
> - **Benchmarking Results.** Conduct extensive benchmarking and detailed performance analysis of 32 leading open-source and proprietary models.
>
> - **Analyses.** Perform extensive analyses such as evaluating model performance across diverse algorithms, detailed reasoning trace analyses, examination of solution submission outcomes, and assessments of model performance under inference-time scaling.

---

> ### Author Response · Authors · 2025-11-27
> **Response to Reviewer bPmJ (2/2)**
>
> **Analysis of Template-based Contamination and Model Performance**
>
> In Appendix F.7, we selected CodeContest [1,2] as our seed dataset to represent template solutions commonly found in training data, as it is among the most prominent competitive coding datasets with over 10,000 problems. For each problem, we sampled three C++ solutions, filtering out solutions shorter than 20 lines or longer than 1,000 lines. Using Dolos [3], we calculated the similarity between official solutions and these template solutions, recording the highest similarity score for each official solution. A higher similarity score indicates a greater likelihood of template-based contamination.
>
> To examine whether template similarity influences model performance, we conducted correlation analyses using GPT-5 and Grok-4-Fast-Reasoning. Our analysis revealed weak correlations between template similarity and relative performance scores: GPT-5 exhibited a Pearson correlation coefficient of 0.136, and Grok-4 demonstrated a Pearson correlation coefficient of 0.193. These correlations, though slightly stronger than those observed in our previous analysis, remain limited.
>
> This finding indicates that template similarity contributes minimally to model performance. Rather, the dominant factor influencing performance is the model's intrinsic problem-solving and reasoning capabilities. Given the nature of competitive coding problems, models rely on learned algorithmic strategies and analytical techniques to solve problems effectively. Therefore, we contend that the minimal template-based contamination observed does not undermine the validity of our evaluations. Models must effectively utilize learned skills from their training, applying them innovatively to solve new and complex challenges.
>
> [1] Li et al., Competition-level code generation with AlphaCode, 2022, Science.
>
> [2] Wang et al., CodeContests+: High-quality test case generation for competitive programming, 2025.
>
> [3] Maertens et al., Discovering and exploring cases of educational source code plagiarism with Dolos, 2024, SoftwareX.

---

### Author Response · Authors · 2025-11-24
**Additional analysis on data contamination (1/2)**

We have included additional analysis of both data contamination and template-based contamination in our benchmark, reflected in the new section 5.4.

**No evidence of temporal performance degradation (5.4.1)**

We evaluate contamination risk by examining whether model performance correlates with problem release dates, particularly around each model’s training cutoff. In Figure A3, our analysis shows no meaningful relationship between publication time and performance: models do not perform better on older tasks, nor do they exhibit any noticeable drop in accuracy on problems released after their knowledge cutoff. This lack of temporal correlation suggests that broad data leakage is unlikely and provides evidence that our benchmark is not driven by memorization effects tied to publication dates.

**Familiarity with task statements and official solutions (5.4.2)**

Following the practice in MLE-Bench[1] on detecting data contamination, we investigated whether the models' familiarity with task statements and official solutions affects their performance. Intuitively, if a model is more familiar with a task, it tends to perform better. Specifically, we define a model’s familiarity with a document as the mean probability it assigns to each token, conditioned on all preceding tokens. Using GPT-OSS-120B, we computed this familiarity metric and plotted it against various performance indicators, including pass rate (Figure A7) and relative score (Figure 5). Our analysis revealed a near-zero correlation between GPT-OSS-120B's familiarity with either task statements or official solutions and its performance. A strong correlation between model performance and familiarity scores could indicate memorization or overfitting to training data. However, our analysis shows almost no correlation between GPT-OSS-120B's familiarity with task statements or official solutions and its performance metrics, suggesting minimal contamination. If significant template-based contamination existed, models highly familiar with official solutions would likely perform substantially better, which we did not observe.

**Low similarity between models' solutions and official solutions (5.4.3)**

Additionally, to investigate whether models rely primarily on memorization of template solutions rather than genuine reasoning, we employ the source code plagiarism detection tool Dolos[2] to measure the similarity between accepted solutions generated by GPT-5 and Grok-4-Fast-Reasoning and the official solutions. GPT-5 achieves a median similarity of 0.11, while Grok-4-Fast-Reasoning has a median similarity of 0.12. In Figure A8, the majority of these solutions exhibit similarity scores below 0.3, strongly indicating that the models do not simply reproduce memorized templates. Instead, the models appear to actively reason and generate novel solutions to solve each problem.

**Negligible correlation between template similarity and model performance differences (Appendix F.4)**

To study if models exhibit similar performance on problems sharing similar solution templates, we compute the official solution similarity for each problem pair and analyze its relationship with the models' performance differences on these problems. Specifically, we assessed GPT-5 and Grok-4-fast-reasoning by plotting the absolute difference in their relative scores against each problem pair's solution similarity (Figure A6). If template-based contamination significantly influenced model performance, we would expect to observe consistently small performance differences between problem pairs with similar official solutions. However, our analysis reveals only a very weak and negligible correlation (GPT-5: Pearson’s r = -0.084; Grok-4: Pearson’s r = -0.072). This indicates that even problem pairs with similar official solution structures often result in substantially different model performances. Additionally, we found that most problem pairs exhibit low similarity, emphasizing the wide variety of problem-solving approaches necessary for success on our benchmark.

**Data Contamination and template-based contamination are minimized in LiveOIBench**

Based on these analyses, we believe that both direct data contamination and template-based contamination are minimized in our benchmark. Moreover, to continuously mitigate potential contamination risks, our benchmark will be regularly updated every 3-6 months with newly released official task statements and high-quality test cases. Consistent with previous studies LiveBench and LiveCodeBench, we define data contamination as the explicit presence of test data within training sets. Furthermore, we argue that even limited template-based contamination does not significantly undermine evaluation validity, since solving these problems effectively demands skill acquisition, generalization, and advanced reasoning abilities from the models.

---

> ### Author Response · Authors · 2025-11-24
> **Additional analysis on data contamination (2/2)**
>
> **Analysis of Template-based Contamination and Model Performance (Appendix F.7)**
>
> In Appendix F.7, we selected CodeContest [3,4] as our seed dataset to represent template solutions commonly found in training data, as it is among the most prominent competitive coding datasets with over 10,000 problems. For each problem, we sampled three C++ solutions, filtering out solutions shorter than 20 lines or longer than 1,000 lines. Using Dolos [2], we calculated the similarity between official solutions and these template solutions, recording the highest similarity score for each official solution. A higher similarity score indicates a greater likelihood of template-based contamination.
>
> To examine whether template similarity influences model performance, we conducted correlation analyses using GPT-5 and Grok-4-Fast-Reasoning. Our analysis revealed weak correlations between template similarity and relative performance scores: GPT-5 exhibited a Pearson correlation coefficient of 0.136, and Grok-4 demonstrated a Pearson correlation coefficient of 0.193. These correlations, though slightly stronger than those observed in our previous analysis, remain limited.
>
> This finding indicates that template similarity contributes minimally to model performance. Rather, the dominant factor influencing performance is the model's intrinsic problem-solving and reasoning capabilities. Given the nature of competitive coding problems, models rely on learned algorithmic strategies and analytical techniques to solve problems effectively. Therefore, we contend that the minimal template-based contamination observed does not undermine the validity of our evaluations. Models must effectively utilize learned skills from their training, applying them innovatively to solve new and complex challenges.
>
> [1] Chan et al., MLE-Bench: Evaluating Machine Learning Agents on Machine Learning Engineering, 2025, ICLR.
>
> [2] Maertens et al., Discovering and exploring cases of educational source code plagiarism with Dolos, 2024, SoftwareX.
>
> [3] Li et al., Competition-level code generation with AlphaCode, 2022, Science.
>
> [4] Wang et al., CodeContests+: High-quality test case generation for competitive programming, 2025.

---

### Author Response · Authors · 2025-11-24
**Performance variation across programming languages**

All evaluations in our benchmark are conducted using C++, as it is the official language for most competitive programming contests and is widely favored in the competitive coding community for its efficiency and performance advantages. Additionally, to provide deeper insights into language-specific model capabilities, we present a comparative analysis (Appendix F.3) of GPT-OSS-120B’s performance across Python, Java, and C++, evaluated on a subset of 132 problems from the USACO contest series.


| Model                  | Relative Score (%) | Pass Rate (%) | Gold (%) | Silver (%) | Bronze (%) | Total Medals (%) |
|------------------------|--------------------|----------------|----------|------------|-------------|-------------------|
| gpt-oss-120b_python    | 44.35              | 46.97          | 13.64     | 13.64       | 27.27        | 54.55             |
| gpt-oss-120b_java      | 50.78              | 56.82          | 9.09      | 36.36       | 27.27        | 72.73             |
| gpt-oss-120b-cpp    | 53.26              | 59.09          | 27.27     | 31.82       | 13.64        | 72.73             |

As shown in the table above, the model performs best when generating solutions in C++, highlighting the advantage of C++ in competitive programming due to its superior execution speed, precise memory control, and optimized functions. Java delivers intermediate performance; although robust and widely supported, it incurs overhead from JVM execution and automated memory management, placing it slightly behind C++. Meanwhile, although Python is the most resourceful and extensively studied language for evaluating LLM coding capability, it shows a considerable performance gap due to its interpreted nature and dynamic typing, sacrificing efficiency.

Similar to Section 5.2, we have analyzed GPT-OSS-120B’s reasoning behaviors across different programming languages. As illustrated in Figure A5, the model consistently allocates reasoning tokens across Python, Java, and C++, indicating that the reasoning strategy remains stable irrespective of language. We hypothesize that the observed performance advantage in C++ primarily stems from its computational efficiency rather than variations in reasoning approach, given the minimal differences in reasoning behavior across these languages.

---

### Author Response · Authors · 2025-12-03
**General Response**

Thank you to all reviewers for their valuable comments and insightful feedback. We have thoroughly addressed all concerns in our revised manuscript. Changes are highlighted in blue and will revert to standard formatting in the final version.

Below is a summary of key updates made during the rebuttal period:

- **Data Contamination:** We have conducted extensive contamination analyses (**Section 5.4, Appendix F.4 and F.7**) demonstrating minimal risk of both data contamination and template contamination.

- **Programming Language Impact:** We have clarified the use of **C++** and provided a comparative analysis with **Java and Python** (**Appendix F.3**).

- **Additional Closed-Source Models:** We have included **Grok-4-Fast-Reasoning** and **Claude-Sonnet-4.5**, with detailed benchmarking results (**Table 2**).

- **Clarification of Overclaims:** In the introduction, we have refined claims to emphasize **LiveOIBench’s unique features** clearly distinguishing it from concurrent benchmarks.

- **Data Quality Assurance:** We have verified high accuracy in **contestant-to-Codeforces profile matching** and **PDF-to-Markdown conversion** (**Appendix B.1 and Appendix B.4**).

- **Hyperparameters and Prompt Setup:** We have provided explicit decoding parameters and evaluation prompt details (**Tables A7, A8, Appendix C**).

- **Prompt Sensitivity:** We have confirmed minimal model performance variation across different prompt structures (**Appendix F.6**).

- **Reasoning Trace Validation:** We have conducted manual validation confirming high accuracy in GPT-OSS-120B's reasoning behavior classification (**Appendix F.9**).

- **Fairness and Consistency:** Besides clarifying our benchmark construction process and its structure, we have also included detailed model performance results across different problem difficulties and contest divisions (**Tables A2, A3, and A9**).

- **Impact of Model Size and Reasoning Tokens:** We have analyzed how model size and reasoning token allocation influence performance across various algorithmic categories (**Appendix F.5**).

**Final Remark**

We appreciate the reviewers' and Area Chair's time and kindly ask you to consider the substantial improvements highlighted in our rebuttal and revised manuscript. We believe LiveOIBench makes a meaningful contribution to the community and hope it receives favorable consideration.

We remain available for further clarifications or questions.

Authors of Submission 23345

---

### Meta-Review · Area_Chair_xgGs · 2026-01-06

**Summary:**

The submission "LiveOIBench: Can Large Language Models Outperform Human Contestants in Informatics Olympiads?" describes a benchmark for coding constructed by scraping all available output-only competition problems from informatics olympiads held across the world since 2022. This results in 403 benchmark questions against which a number of LLMs are evaluated.
On the plus side, the authors do good work to collect human ratings and provide a good comparison to human contestant.

**Reviewer Concerns:**

There are several concerns with this paper based on which I do not recommend accepting it in its current state.
1. *Data Contamination.* This issue is brought up by a number of reviewers. The submission starts out by arguing that "To mitigate potential contamination from pre-training datasets, we strictly limit our dataset to contests held after 2022", this sentence is immediately incorrect, as only models with a cutoff after 2022 are tested. Only when pressed by reviewers does the author response during the rebuttal period argues that contamination may not be a problem with current models, by showing that current solutions may be different from reference solutions where available, and by showing that performance between years does not show a strong trend (however performance between years varies greatly due to the small pool of questions).

    But, conceptually, there is no mechanism used in this work to make this a contamination-free benchmark. Prior work, such as IO-Bench spends significant effort on this, deliberately constructing a private test set, and other prior work carefully constructs decontamination pipelines and canaries that would reveal contamination, so there are a number of mechanisms in the literature that are only an afterthought here.

2. *Is this a live benchmark*? In what sense is this a live benchmark? This statement appears predicated on the submission statement to
    > Updates with newly released Olympiad tasks
maintain benchmark freshness and minimize data contamination risks, supporting continuous
monitoring of LLM coding capabilities on challenging programming problems.

     But, this is only a claim about future updates that could have been included in each of the papers discussed in the related work section as offline. Mechanically, this seems unsupported. Parts of the pipeline require manual intervention, Appendix B.1:
     > To accomplish this, we manually copy the raw webpage content into Gemini-2.0-Pro and prompt it to generate CSV files with normalized header

   and from the *missing data* section

    > For contests missing test cases on the official site, we employ a parser to retrieve them from CSES [third-party website] and integrate them into our dataset.

     And the paper does not construct its analysis in a form or discuss how the scores should be evaluated or denoted once new data comes in, aside from using a slider to select a date range on the website. The repository for the supplementary material provided at https://anonymous.4open.science/r/LiveOIBench-25F9/ is empty so I also could not exonerate the submission by checking whether these questions have been considered and not included in the submitted draft.

3. *Expert-designed private test cases*? The submission advertises that its test cases are expert-designed and private. But, aside from the introduction, I can find no information on this. Based on my careful read of the paper, this statement refers simply to the test cases scraped from the olympiad webpages (Appendix B.1)? The test cases are technically expert-written, given that they are competition material, but the submission overstates this, and only 10% of tasks were even manually checked. The submission only verifies that reference solutions provided by the competition do indeed pass these test cases. Further, I do not understand in what way these competition test cases are private?

 **Regarding reviewer comments.**

Reviewer bPmj notes as a strength that
> The Continuous Updates of LiveOIBench are achieved through five core mechanisms: dynamic task injection, user feedback-driven iteration, versioned management, scoring rule calibration, and delayed release strategy. Its primary goals are to maintain the benchmark’s timeliness, discriminative power, and fairness, while combating data contamination and model overfitting. This design paradigm aligns closely with cutting-edge dynamic benchmarks such as LiveBench and LiveCodeBench, reflecting the latest trends in the current AI evaluation field.

But this is simply not a statement about the submission at hand? "dynamic task injection", " user feedback-driven iteration"? "Versioned management" would be a good concept to apply here about this does not appear in the submitted draft.

The reviewer's main question is
> Is a Serendipity Metric—designed to calibrate template similarities—necessary to make the benchmark more contamination-free?

But, 'calibrate template similarities' is nonsensical.

Reviewer rxTf only notes that as a strength that they like the webpage particularly compelling, but the webpage is a straightforward table just like the LiveCodeBench website. Also, I could not find any evidence that the submitted draft does "extensive work to setup a website that will pull new challenging problems in from future coding contests".

Based on concerns with these reviews as illustrated above I have chosen to instead mainly review the paper directly, as shown above.


RVy3 and faYZ note smaller concerns regarding hyperparameter details, language choice (but all olympiads are done in C++) and reproducibility of the model-based evaluation of reasoning traces which the authors answer adequately.

---

Aside from content concerns as discussed above, reading the submission I felt that it was continuously very strongly worded and written to be maximally convincing in a way that I found disconcerting, and in disconnect with the concrete content of the submission.

**Reviewer Scores:**

no changes expected.

---

### Decision · Program_Chairs · 2026-01-26

Reject